# MOBILELLM-R1:
# EXPLORING THE LIMITS OF SUB-BILLION LANGUAGE MODEL REASONERS WITH OPEN TRAINING RECIPES

**Changsheng Zhao**[*]     **Ernie Chang**[* §]     **Zechun Liu**[* †]
**Chia-Jung Chang**     **Wei Wen**     **Chen Lai**     **Sheng Cao**     **Yuandong Tian**
**Raghuraman Krishnamoorthi**     **Yangyang Shi**     **Vikas Chandra**
Meta AI
{cszhao, erniecyc, zechunliu}@meta.com

## ABSTRACT

The paradigm shift in large language models (LLMs) from instinctive responses to chain-of-thought (CoT) reasoning has fueled two prevailing assumptions: (1) reasoning capabilities only emerge in sufficiently large models, and (2) such capabilities require training on massive datasets. While the first assumption has already been challenged by recent sub-billion-parameter reasoning models such as Qwen3-0.6B and DeepSeek distilled variants, the second remains largely unquestioned. In this work, we revisit the necessity of scaling to extremely large corpora (>10T tokens) for reasoning emergence. By carefully curating and resampling open-source datasets that we identify as beneficial under our designed metrics, we demonstrate that strong reasoning abilities can emerge with far less data. Specifically, we show that only ∼2T tokens of high-quality data are sufficient, and pre-training with 4.2T tokens on the dataset resampled from these ∼2T tokens, followed by a established post-training procedure, enables the development of MobileLLM-R1, a series of sub-billion-parameter reasoning models that substantially outperform prior models trained on fully open-sourced data. For example, MobileLLM-R1-950M achieves an AIME score of 15.5, compared to just 0.6 for OLMo-2-1.48B and 0.3 for SmolLM-2-1.7B. Remarkably, despite being trained on only 11.7% of the tokens compared to Qwen3's proprietary 36T-token corpus for pretraining, MobileLLM-R1-950M matches or surpasses Qwen3-0.6B across multiple reasoning benchmarks. To facilitate further research in this direction, we have made the models and code publicly available, along with the complete training recipe, data sources, and data mixing ratios.

## 1 INTRODUCTION

Large language models (LLMs) such as GPT (Achiam et al., 2023), Qwen (Yang et al., 2025; 2024), and DeepSeek (Guo et al., 2025) have demonstrated remarkable progress in explicit reasoning. Advances have been driven by scaling model size, expanding training data, and applying post-training techniques such as supervised fine-tuning (SFT) and reinforcement learning (RL). Reasoning LLMs are capable of tackling complex problems by following long chains of thought that incorporate reflection, backtracking, and self-validation. At the same time, reasoning traces have evolved

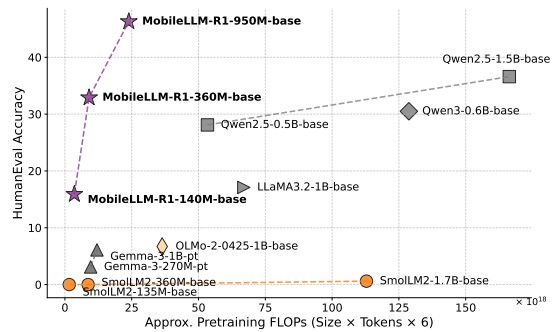

Figure 1: Pretrained model accuracy vs. training efficiency trade-off.

---

[*]Equal contribution, authors listed in alphabetical order. [§]Led overall data curation efforts
[†]Corresponding author

from prompt-based chain-of-thought (CoT) in-context learning (Wei et al., 2022) to models explicitly optimized on long reasoning traces to generate multi-step reasoning sequence (Jaech et al., 2024).

However, this paradigm poses increasing challenges for real-world deployment. Large models already strain resource-constrained devices (Liu et al., 2024), and long-context reasoning further exacerbates memory usage as KV cache growth sharply increases the footprint (Sadhukhan et al., 2025). Looking ahead, one can envision a future with personal assistants, smart homes, and robots increasingly relying on on-device reasoning for complex tasks. In such a world, deployability and portability will become inevitable trends for the next generation of LLMs. This motivates our central question: *Given strict capacity constraints, what is the most effective recipe to endow small reasoning models with strong capabilities and unlock their hidden potential?*

Developing small reasoning models poses unique challenges beyond simply scaling down large ones. For large models, expanding the corpus often drives stronger generalization. In contrast, small language models are far more sensitive: noise in the data can easily overwhelm their limited capacity, making data quality and curation paramount. As models shrink, neurons must encode more overlapping knowledge, increasing the risk of interference and conflicts (Zhu et al., 2025)—superposition provides an intuitive lens for understanding this challenge (Elhage et al., 2022). Mitigating these risks requires carefully optimized data, objectives, and training procedures.

While extensive research has explored how post-training objectives and data curation can elicit reasoning from pretrained models (Wang et al., 2025; Li et al., 2025), far less attention has been paid to a more fundamental question: *How can we endow pretrained models with the latent potential for reasoning in the first place?* This work addresses this gap by investigating two critical questions: (1) What kinds of data are most effective for instilling reasoning capability, and (2) How can diverse forms of reasoning—such as coding, mathematics, and logical problem-solving—be embedded into a compact model without overwhelming its limited capacity?

Through capability-aware data curation and probing into the latent factors that govern reasoning, we achieve highly token-efficient pretraining compared to prior work. With only 4.2T training tokens, just 11.7% of Qwen's 36T, our `MobileLLM-R1`-950M model, matches or surpasses Qwen3-0.6B Yang et al. (2025) on multiple reasoning benchmarks, placing itself on the Pareto frontier of accuracy–training-token efficiency trade-off curve (Figure 1). Beyond introducing a high-performing small-scale reasoning model, we share both the insights and the pitfalls encountered along the way, offering a first-hand glimpse into the complex yet fascinating mechanisms behind reasoning models.

Our contributions are as follows:

- We introduce benchmark-free, self-evolving data optimization for pre-training data curation, a principled dataset-level weighting approach that leverages cross-domain influences to tailor the data mixture. This facilitates robust reasoning generalization on held-out benchmarks, achieved without exposing them during training or data mixture optimization.

- We further propose a data–model co-evolution strategy to adapt to rapid changes in model capacity during mid-training. We show that this process converges as most samples reach zero or negative influence, indicating that the dataset's information has been largely exhausted and offers minimal further improvement.

- Compared to existing fully open-source models, `MobileLLM-R1`-950M model attains $5\times$ higher MATH accuracy than Olmo 1.24B (Allal et al., 2025) and $2\times$ higher than SmolLM2 1.7B (Allal et al., 2025), while significantly outperforming both on code benchmarks despite having fewer parameters.

- We have disclosed the complete set of open-sourced datasets employed in our study and have released all trained models and accompanying code to enable full reproducibility and foster future research.

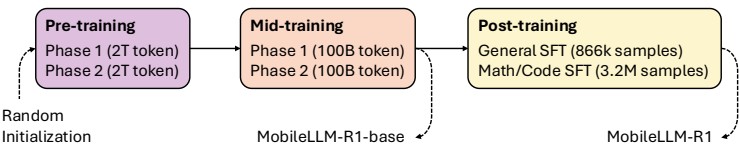

Figure 2: Overall training pipeline of `MobileLLM-R1`.

## 2 PRE-TRAINING: BALANCE OF CAPABILITIES

The notion of *reasoning* in large language models (LLMs) remains both complex and contested. While the term is sometimes used to describe a model's ability to engage in structured, multi-step inference (Wei et al., 2022; Kojima et al., 2022), it has also become a proxy for improved performance on challenging benchmarks (Srivastava et al., 2022). In this work, we adopt a pragmatic stance: we treat gains on reasoning-centric benchmarks as reasonable evidence of enhanced reasoning *behaviors*, while remaining cautious about equating such gains with genuine reasoning ability in the cognitive sense (Bender & Koller, 2020). Concretely, this entails selecting informative datasets that most effectively enhance the target capability (Section 2.1) and optimizing their combination ratios to maximize knowledge acquisition within the fixed token budget (Section 2.2). Figure 2 illustrates the training pipeline with the full procedure in Appendix A.

### 2.1 SELECTING INFORMATIVE DATASETS FOR TARGET CAPABILITY

To systematically assess which pre-training distributions most effectively support downstream reasoning behaviors, a naïve approach would be to pre-train separate models on all combinations of candidate datasets, followed by mid-training and post-training, and then measure performance on reasoning benchmarks. However, this strategy is both computationally prohibitive and prone to overfitting to specific benchmarks.

Instead, we design a leave-one-out (LOO) analysis. We train models from scratch on the entire set of pre-selected high-quality datasets, excluding one dataset at a time. We then trace negative log-likelihood (NLL) on curated *capability-probing datasets* throughout training. Each *capability-probing dataset* can be viewed as defining a token distribution that implicitly induces the necessary preconditions for reasoning to emerge. Importantly, these distributions are heterogeneous: when learned, they contribute unequally to different reasoning-related capabilities, such as *code understanding*, *general knowledge*, and *mathematical problem solving* (Chen et al., 2021; Hendrycks et al., 2021; Cobbe et al., 2021).

#### 2.1.1 CURATION OF REPRESENTATIVE DATASETS AND CAPABILITY-PROBING DATASET

Curating the *capability-probing datasets* is critical: it must be representative of the desired capabilities and sufficiently comprehensive to cover each reasoning category. We describe the process of preparing capability datasets as follows.

**Hierarchical Rejection Sampling.** To derive a compact *capability-probing datasets* for each domain, we employ a hierarchical rejection sampling pipeline that integrates multiple classifier- and model-based filters. The objective is to construct a small yet representative target dataset for each capability, such that it can serve as a faithful proxy for reasoning performance while dramatically reducing overall volume during evaluation. For each corpus in Table 5, we first apply the FINEWEB-EDU classifier (Penedo et al., 2024) to select samples with high educational value, retaining only those with classifier scores above 4. Next, we incorporate model-based evaluation by scoring each remaining sample using the Ask-LLM paradigm (Sachdeva et al., 2024). The evaluation prompt asks the model to judge whether a sample should be included in a reasoning-probing dataset, framed as a binary classification task ("1" for inclusion, "0" for exclusion). Rather than relying solely on the hard prediction, we record the probability assigned to "1" as a graded measure of the model's confidence in the example's reasoning relevance. For all Ask-LLM scoring, we select the top 10% samples within each dataset. This step complements classifier-based quality filtering by directly capturing signals of reasoning relevance, consistent with recent findings that costly, fine-grained quality samplers can outperform simple maximum-coverage approaches in terms of data efficiency (Sachdeva et al., 2024; Pang et al., 2025; Chen & Zhou, 2025). Next, we apply a domain-specific prompt to Ask-LLM for each capability with specific emphasis on code, math, general knowledge or combined. Finally, we perform semantic deduplication across corpora, shrinking each dataset in Table 5 to a subset of roughly 10,000 examples. This yields the ***representative datasets*** $\mathcal{D}^{\mathcal{R}}_i$, each containing highly representative samples for its corresponding corpus.

We categorize them into three domains according to their composition: Code ($\mathcal{C}$), Math ($\mathcal{M}$), and Knowledge ($\mathcal{K}$):

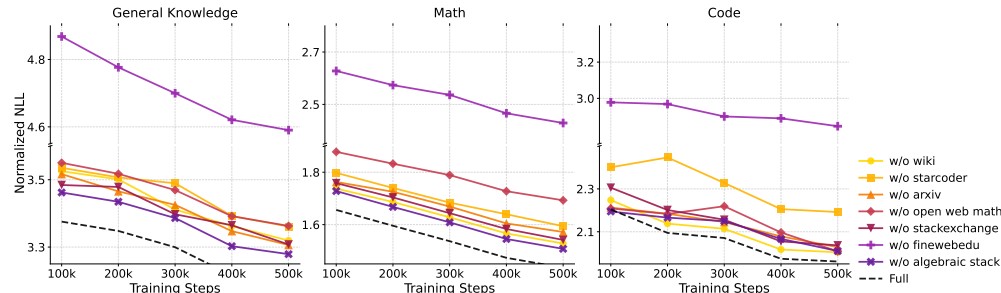

Figure 3: Leave-one-out analysis of pretraining data. The y-axis represents the normalized negative log-likelihood (NLL) on the curated *capability probing datasets*. We systematically exclude individual datasets—StarCoder, OpenWebMath, FineWeb-Edu, Wiki, Arxiv, StackExchange, and Algebraic Stack – to quantify their per-token contributions to downstream performance by comparing them with using full set of data. These trajectories reveal how beneficial each dataset is and how its impact evolves throughout training. Removing FineWeb-Edu yields the largest cross-domain degradation, likely attributable to its board web-based composition that connects diverse domain.

- $\mathcal{C}$ = {StarCoder, StackExchange, Nemotron-Code, Cosmopedia, Natural Reasoning, pes2o}
- $\mathcal{M}$ = {OpenWebMath, FineMath, Algebraic Stack, Nemotron-Math, Cosmopedia, Natural Reasoning, pes2o}
- $\mathcal{K}$ = {FineWeb-Edu, Wikipedia, Arxiv, Cosmopedia, Nemotron-Science, Natural Reasoning, pes2o}

Note that a single dataset may contain data relevant to multiple domains, in which case its representative subset is included in more than one domain. In this way, we construct three filtered, domain-specialized *capability-probing datasets*, $\mathcal{D}_{\mathcal{C},\mathcal{M},\mathcal{K}}^{\mathcal{P}}$, by combining the representative subsets from all datasets assigned to each domain. We use $(\mathcal{C},\mathcal{M},\mathcal{K})$ to denote a *mixture of original datasets* prior to down-sampling.

### 2.1.2 DISENTANGLING THE IMPACT OF DATA SOURCES

We then evaluate the impact of different pretraining corpora on the emergence of reasoning ability by measuring the negative log-likelihood (NLL) on the *capability-probing datasets*. To isolate the contribution of each corpus, we perform rigorous leave-one-out ablation studies, systematically removing individual datasets and measuring the resulting change in NLL across the three *capability-probing datasets* corresponding to *Code*, *Math*, and *General Knowledge* capabilities.

**Group Impact via Loss Delta.** We define the impact of a dataset $\mathcal{D}_j$ on a reasoning capability as the change in loss it induces on the corresponding *capability-probing dataset* $\mathcal{D}_{\mathcal{C},\mathcal{M},\mathcal{K}}^{\mathcal{P}}$. Let $\hat{\theta}$ denote parameters trained on the full dataset $\mathcal{D} = \cup_i \mathcal{D}_i$, and $\hat{\theta}_{-j}$ denote parameters trained with $\mathcal{D}_j$ removed. The *group impact* of $\mathcal{D}_j$ on $\mathcal{D}_c^{\mathcal{P}}, c \in \{\mathcal{C},\mathcal{M},\mathcal{K}\}$ is

$$\Delta\mathcal{L}(\mathcal{D}_j, \mathcal{D}_{\mathcal{C},\mathcal{M},\mathcal{K}}^{\mathcal{P}}) = \mathbb{E}_{z\sim\mathcal{D}_{\mathcal{C},\mathcal{M},\mathcal{K}}^{\mathcal{P}}}\big[\ell(z; \hat{\theta}_{-j}) - \ell(z; \hat{\theta})\big], \tag{1}$$

where $\ell$ is the evaluation loss. A positive value indicates that removing $\mathcal{D}_j$ increases the benchmark loss (i.e., $\mathcal{D}_j$ is beneficial), while a negative value suggests that its presence may hurt performance.

**Leave-One-Out Ablations.** We operationalize Eq. 1 by training models under leave-one-out settings and measuring the resulting differences in loss across benchmarks. Together, these analyses highlight not only *which* sources matter most, but also *how much* marginal benefit additional data from a given source provides. This methodology allows us to disentangle the contributions of heterogeneous data sources to reasoning-related performance in code, knowledge, and mathematics.

Figure 3 presents the results of our leave-one-out (LOO) experiments across the three evaluated capabilities. To ensure fairness, tokens from each dataset are sampled with equal probability, and no example is repeated during pretraining. Without this normalization, larger datasets such as FINEWEB-EDU would otherwise dominate exposure. We find that excluding FINEWEB-EDU results in the largest degradation across all capabilities, including knowledge, math, and code. We attribute this to its web-based composition, which provides broad and diverse coverage across domains. This

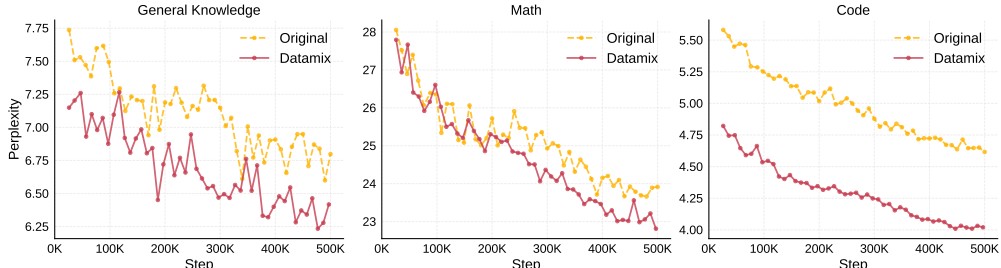

Figure 4: Comparison of data mixture strategies using averaged perplexity. Math is averaged over MATH-500, GSM8K, Code on HumanEval, and General Reasoning is an average of 9 tasks, including ARC-easy, ARC-challenge, BoolQ, PIQA, SIQA, HellaSwag, OBQA, WinoGrand, and MMLU. We compare **Original** (uniform) sampling with our derived **Datamix**. This mixture consistently lowers PPL on Code, Math, and Knowledge benchmarks, despite these benchmarks not being used during training or data selection.

result highlights the central role of large-scale web data as a form of "glue" that binds heterogeneous domains together.

In contrast, domain-specific datasets primarily strengthen their respective domains: STARCODER substantially improves code performance (and, interestingly, math), while math-focused corpora primarily benefit math. However, their transfer to general knowledge is limited. An unexpected observation is that STARCODER benefits math more than OPENWEB-MATH benefits code, a reversal of the commonly held view that mathematical data contributes disproportionately to coding ability Lewkowycz et al. (2022). Finally, WIKIPEDIA appears to contribute little to math or code compared to web or domain-specific data, yet remains necessary as a structured and reliable source of factual knowledge.

## 2.2 DATAMIXING VIA CROSS-CAPABILITY SELF-INFLUENCE

In Section 2.1.2, we demonstrate that the pre-selected datasets yield measurable utility, as evidenced by reductions in NLL on *capability-probing datasets*. Building on this, we study token budget allocation: given a fixed training budget, how should tokens be distributed across heterogeneous datasets to maximize downstream reasoning performance? Uniform sampling provides a natural baseline but ignores the varying marginal utility of different datasets. Our key insight is that more informative datasets should receive proportionally larger sampling ratios. To operationalize this, we leverage the *influence score* to quantify each dataset's contribution and guide principled token re-weighting.

Generally, let $\theta^*$ denote parameters obtained by training on dataset $\mathcal{D}$, $x_i$ a training example, $x_{\text{test}}$ a example from the target set, which in our case is *capability probing set*, and $\mathcal{L}(x, \theta)$ the loss function. The *influence score* of $x_i$ on the test loss can be approximated as

$$I(x_i, x_{\text{test}}; \theta) = -\nabla_\theta \mathcal{L}(x_{\text{test}}; \theta^*)^\top H_{\theta^*}^{-1} \nabla_\theta \mathcal{L}(x_i; \theta^*), \tag{2}$$

where $H_{\theta^*}$ is the Hessian of the training loss at $\theta^*$. While directly computing the Hessian matrix $H_{\theta^*}$ is computationally prohibitive for large models, *AutoMixer* (Chang et al., 2025) proposes an efficient approximation method that bypasses explicit Hessian inversion and makes influence score calculation scalable.

We extend the *AutoMixer* framework by treating influence scores as quantitative proxies linking individual training samples to capabilities. Concretely, the influence of a sample on the validation loss of a capability-probing dataset measures the *connection strength* between the sample and the corresponding capability. Rather than using benchmark test sets, we employ samples from *capability-probing datasets* and compute influence scores separately for Code ($\mathcal{C}$), Math ($\mathcal{M}$), and Knowledge ($\mathcal{K}$) domains.

For each training sample $x_i$ from a source dataset, we compute its influence on the validation loss of all three capability-probing datasets. We term this "self-influence" when training and validation samples originate from the same capability and "cross-influence" if they target different capabilities. Because the source datasets are substantially large, we develop an efficient influence estimation algorithm that operates on the *representative dataset* (defined in Section 2.1.1) of each source in Table 5, yielding a computationally scalable surrogate that faithfully preserves cross-capability

contribution signals. Concretely, if $x_i \in \mathcal{D}^{\mathcal{R}}{}_{\text{StarCoder}} \subset \mathcal{D}^{\mathcal{P}}_{\mathcal{C}}$, we evaluate

$$\text{Self-influence: } \mathcal{I}(x_i, x_{test} \in \mathcal{D}^{\mathcal{P}}_{\mathcal{C}}; \theta_{\mathcal{C},t}), \tag{3}$$

$$\text{Cross-influence: } \mathcal{I}(x_i, x_{test} \in \mathcal{D}^{\mathcal{P}}_{\mathcal{M}}; \theta_{\mathcal{M},t}), \ \mathcal{I}(x_i, x_{test} \in \mathcal{D}^{\mathcal{P}}_{\mathcal{K}}; \theta_{\mathcal{K},t}) \tag{4}$$

Here, checkpoints $\theta_{\mathcal{C},t}$, $\theta_{\mathcal{M},t}$, and $\theta_{\mathcal{K},t}$ are obtained by training separate models to convergence on the full training sets of domains $\mathcal{C}, \mathcal{M}, \mathcal{K}$, yielding domain-specialized parameters. Following the *AutoMixer* protocol, a single checkpoint is insufficient to capture the full training dynamics. We therefore compute influence scores at $T = 10$ evenly spaced checkpoints, weighting each score proportionally to its training step to emphasize later-stage training. These weighted scores quantify the evolving influence of example $x_i$ on the Code, Math, and Knowledge domains throughout training.

Then, the joint influence of a sample is computed as

$$\mathcal{I}_{\text{joint}}(x_i) = \sum_{c \in \{\mathcal{C}, \mathcal{M}, \mathcal{K}\}} \sum_{t=1}^{T} \alpha_{c,t} \cdot \mathcal{I}(x_i; \theta_{c,t}), \tag{5}$$

where $\theta_{c,t}$ is the checkpoint $t$ for capability $c$, and $\alpha_{c,t}$ are blending factors reflecting acquisition speed across checkpoints. We assign linearly increasing weights $\alpha_{c,t} \propto t$ across the $T$ checkpoints, and maintain uniform weights across capabilities $c$.

Each source dataset $g$ is then assigned a sampling weight $(w_g)$:

$$w_g = \frac{\rho_g}{\sum_{g'} \rho_{g'}}, \quad \rho_g = \frac{1}{N_g} \sum_{x_i \in g} \mathcal{I}_{\text{joint}}(x_i) \cdot s_i, \tag{6}$$

with $N_g$ the token count of dataset $g$ and $s_i$ the length of sample $x_i$. The resulting mixture respects the global budget $N$ while prioritizing datasets whose samples show strong self- and cross-capability connections.

In this setup, we derive a closed-form solution for the data mixture ratio, enabling effective utilization of the limited token budget while enhancing each dataset's contribution to model performance. Using the *representative datasets* and *capability probing datasets* sampled from the training corpus, it makes influence score computation tractable and exposes how strongly each source dataset (Table 5) contributes to Code, Math, and Knowledge capabilities. This formulation enables principled weighting at the *dataset level*, grounding the mixture in empirically measured cross-domain influences rather than heuristic allocation. As illustrated in Figure 4, the resulting mixture consistently outperforms uniform sampling on Code, Math, and Knowledge benchmarks—none of which are accessed during training or mixture construction—demonstrating the potential for benchmark-free, self-adaptive data optimization.

## 3 MID-TRAINING: KNOWLEDGE COMPRESSION

After the model has been exposed to broad knowledge during pretraining, the mid-training phase focuses on compressing this knowledge and maximizing performance on target tasks. We design each mid-training phase with a limited budget of 100B tokens. Unlike pretraining, mid-training induces dramatic shifts in weight distributions and necessitates a more sophisticated, co-evolving model–data mixture strategy. To this end, we propose a novel mid-training paradigm that enables self-boosting: the model trained on a given data mixture is used to compute influence scores for samples, which are then leveraged to dynamically remove negative influence samples and adjust the data sampling ratios for the next phase. As training progresses, the influence scores of data samples increasingly concentrate around zero or negative values, indicating near-complete utilization of the informative content in the dataset and convergence of the process. Notably, this self-evolving scheme requires no access to external benchmark datasets, yet it substantially improves performance on target benchmarks relative to uniform sampling.

We build upon the Dolmino dataset, which has been shown in the OLMo 2 (OLMo et al., 2024) to be an effective mid-training corpus. To enhance domain specialization, we augment Dolmino with additional mathematics and programming data, aiming to strengthen the model's math and coding capabilities. Given a training example $x_i$ from the mid-training dataset and a probe example $x_{\text{test}}$ from *capability probing dataset* $\mathcal{D}^{\mathcal{P}}_{\mathcal{C},\mathcal{M},\mathcal{K}}$, we calculate the influence score $\mathcal{I}(x_i, x_{\text{test}}; \theta)$. Here, rather than relying on separately trained models for domain-specific corpora, we leverage the pretrained model

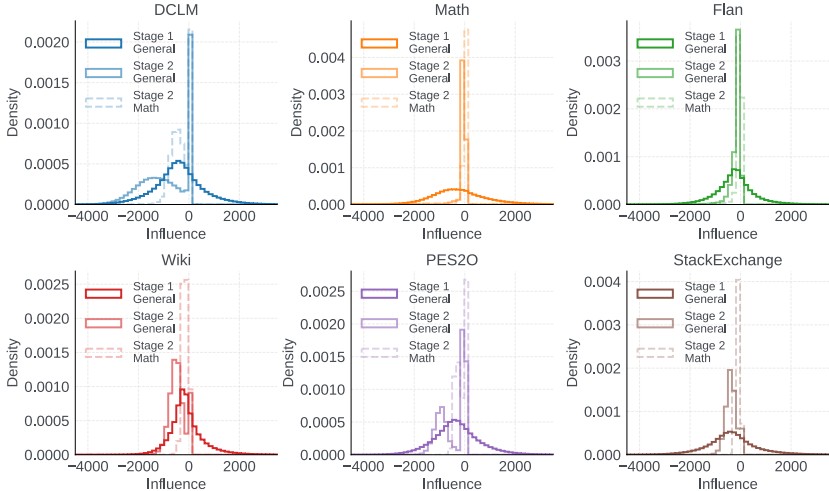

Figure 5: Histogram of influence scores for the general knowledge and math capabilities. In phase 1, data samples exhibit varied influence scores. As training progresses, most samples eventually attain zero or negative influence scores, indicating a convergence point in the model–data co-evolution. At this stage, the dataset's information has been largely exhausted and can no longer contribute to further model improvement.

$\theta$ to capture the dataset requirements at the current stage of training. The data–model co-evolution proceeds iteratively through the following steps:

(1) *Sample-level influence for rejection sampling.* Intuitively, this step acts as a filtering mechanism: only training examples that positively contribute to the target capabilities are retained, while neutral or detrimental samples are discarded. Given the raw mid-training dataset $\mathcal{D}^{(\mathrm{raw})}$, at compression phase $t$ we define the retained dataset as:

$$\mathcal{D}_t = \{x_i \in \mathcal{D}^{(\mathrm{raw})} : I(x_i; \theta_t) > 0\}, \tag{7}$$

where $\theta_t$ is the model state at phase $t$. This rejection sampling can be interpreted as an iterative data distillation process: the model continually refines its training distribution, focusing only on samples that yield positive transfer toward the target probing dataset.

(2) *Dataset-level influence for adaptive data mixing.* Beyond sample-level filtering, we aggregate influence scores to the dataset level, enabling adaptive control of the mixing ratio among mid-training datasets, according to Eqs. 5 and 6).

(3) *Train the model on the curated data and repeat the iterative process.* The compressed dataset with the updated mix ratio is used for continued mid-training:

$$\theta_{t+1} = \mathrm{MidTrain}(\theta_t, \mathcal{D}_t). \tag{8}$$

and the updated model $\theta_{t+1}$ provides refined influence scores for the next stage. This iterative compression continues until no additional samples yield a positive influence score. In practice, we find that two stages suffice to produce a well-compressed dataset that balances generality with targeted capability improvements.

**Intuition: Distributional Compression of Influence.** The compression phases can be viewed as iteratively distilling the mid-training dataset in alignment with the model's evolving capacity throughout training. In early phases, the influence scores are more varied because the model $\theta_t$ is still under-trained. However, as $t$ increases, the model becomes better aligned with the target distribution, and its estimates of sample importance are narrowed down (See Figure 5). This recursive interplay produces increasingly refined datasets: uninformative (or negative-influence) samples are discarded, thus amplifying the impact of high-value samples.

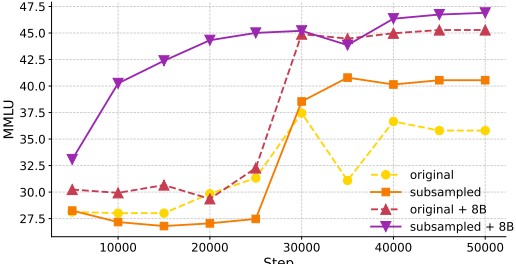

Figure 6: Comparison of the impact on the MMLU benchmark between the original mid-training data and the subsampled data, with and without knowledge distillation.

Conceptually, compression phases mimic an iterative denoising process, where each step sharpens the signal from $\mathcal{D}^{(\text{target})}$ against the noisy background of $\mathcal{D}^{(\text{raw})}$. We terminate the iteration until the distribution of influence converges to approximately zero.

Figure 5 shows histograms of influence scores for general knowledge and math across stage 1 and stage 2 of training. During stage 2, the distribution of influence scores undergoes a pronounced "compression": the range of values narrows, and extreme contributions become less pronounced. Intuitively, as the model becomes more capable, the influence of individual data samples converges toward zero, indicating diminishing impact on downstream reasoning performance. We further highlight the effect of this influence compression in Figure 6. Subsampled mid-training data consistently outperforms the original mid-training set under both standard cross-entropy training and knowledge distillation. Notably, the original data experiences a pronounced performance dip around 30K steps, whereas the subsampled data maintains higher downstream performance throughout training. A similar trend occurs with knowledge distillation using the LLaMa3-8B teacher model, though the performance gap is slightly smaller than under pure cross-entropy. These results indicate that compressing influence scores effectively identifies and preserves the most informative samples, leading to more robust and stable performance trends.

## 4 EXPERIMENTAL RESULTS

Using the datasets from Sections 2 and 3, we obtain `MobileLLM-R1`-base. Given that our primary goal is to elucidate how data curation in pre-training and mid-training builds strong small reasoning models, we leverage established supervised fine-tuning (SFT) datasets. We first apply Tülu-3-SFT (Lambert et al., 2024) dataset for instruction alignment and OpenScienceReasoning-2, OpenCodeReasoning-2 (Ahmad et al., 2025) and OpenMathReasoning (Moshkov et al., 2025) for reasoning-oriented SFT to extend context and elicit long chain-of-thought reasoning. We validate our training process and data choices, compare our model against prior work trained on the same reasoning SFT datasets.

**Post-training process ablation:** Our ablation studies (Table 1) reveal several key insights into the two-stage post-training pipeline. (1) Instruction-following supervision (Tulu-SFT) provides crucial alignment signals that make subsequent reasoning adaptation significantly more effective than starting directly with reasoning data. (2) Domain-specific reasoning corpora (math, science, code) yield consistent gains on their respective benchmarks, while scientific reasoning data further exhibits strong cross-domain transfer to math and code. (3) Symbolic reasoning improvements often trade off with factual knowledge retention, as introducing math or code data reduces MMLU performance, particularly in smaller models with limited capacity. (4) Decoupling alignment and reasoning proves essential: a staged approach (Tulu first, then reasoning data) consistently outperforms joint training, especially on math and general reasoning benchmarks.

**Comparison with baselines on identical reasoning SFT:** To disentangle the contribution of curated pre-training and mid-training data from that of high-quality post-training data, we conducted an ablation study. Specifically, we finetune all baseline instruct models, as well as the `MobileLLM-R1`

Table 1: Ablation studies on post-training stages. We use M, S, C to denote OpenMathReasoning, OpenScienceReasoning-2 and OpenCodeReasoning-2 datasets respectively.

| Stage 1 | Stage 2 | MATH | GSM8K | LCBv6 | MMLU |
|---|---|---|---|---|---|
| *Ablation on Stage 1* | | | | | |
| w/o Tulu-3 | M + C + S | 56.2 | 68.2 | 13.1 | **44.0** |
| w/ Tulu-3 | M + C + S | **57.8** | **68.5** | **13.7** | 43.7 |
| *Ablation on Stage 2* | | | | | |
| Tulu-3 | M | 57.4 | 68.2 | 0.0 | 43.1 |
| Tulu-3 | C | 16.2 | 31.0 | 12.0 | 39.9 |
| Tulu-3 | S | 23.8 | 62.2 | 3.4 | **45.6** |
| Tulu-3 | M + C | 58.4 | 65.6 | 10.9 | 40.4 |
| Tulu-3 | M + S | **60.0** | 66.9 | 0.6 | 45.0 |
| Tulu-3 | C + S | 29.4 | 65.3 | **14.3** | 44.4 |
| Tulu-3 | M + C + S | 57.8 | **68.5** | 13.7 | 43.7 |
| *Joint Ablation* | | | | | |
| Tulu-3 + (M + C + S) | – | 56.2 | 53.1 | **14.9** | **44.0** |
| Tulu-3 | (M + C + S) | **57.8** | **68.5** | 13.7 | **44.0** |

Table 2: Evaluation of reasoning capabilities elicited by different models when fine-tuned on the same reasoning supervised fine-tuning (SFT) dataset. Baseline models use their instruct checkpoints; our model uses intermediate Tulu3-SFT checkpoints, denoted with *. All models are trained for one epoch on the joint reasoning SFT corpus (OpenMathReasoning + OpenScienceReasoning-2 + OpenCodeReasoning-2).

| Model | Size | MATH | GSM8K | LCBv6 |
|---|---|---|---|---|
| SmolLM2-135M-Instruct | 135M | 3.2 | 1.6 | 0.6 |
| **MobileLLM-R1-140M*** | 140M | **4.8** | **3.7** | **1.1** |
| SmolLM2-360M-Instruct | 362M | 5.2 | 7.4 | 3.4 |
| **MobileLLM-R1-360M*** | 359M | **19.2** | **23.8** | **4.0** |
| OLMo-2-0425-1B-SFT | 1.48B | 53.0 | 58.8 | 11.4 |
| SmolLM2-1.7B-Instruct | 1.71B | 41.4 | 50.5 | 7.4 |
| **MobileLLM-R1-950M*** | 949M | **57.8** | **68.5** | **13.7** |

general supervised fine-tuned model (trained for 2 epochs on the Tulu dataset), on the joint reasoning SFT corpus (OpenMathReasoning + OpenScienceReasoning-2 + OpenCodeReasoning-2) for one epoch. Our results in Table 2 show that, even under identical supervised fine-tuning, models with stronger pre-training and mid-training exhibit more robustly embedded knowledge, which in turn facilitates the elicitation of reasoning capabilities during post-training. In that sense, `MobileLLM-R1` consistently outperforms prior models trained on fully open-source corpora, such as OLMo-2 and SmolLM, on reasoning benchmarks. Notably, our 140M and 360M checkpoints achieve substantial gains over SmolLM baselines, while our 950M model surpasses both OLMo-2 1.48B and SmolLM-1.7B, despite its significantly smaller size.

## 4.1 FINAL RESULTS

While the model undergoes the pre-training, mid-training, and post-training stages, we track its reasoning ability by measuring perplexity on two reasoning-focused benchmarks: HumanEval and GSM8K. Our results in Figure 7 show that the perplexity on math sees a significant drop early during the second phase of pre-training. Interestingly, the same model, when subjected to the second phase of mid-training with limited data, exhibits a dramatic perplexity decrease in HumanEval. This suggests that the knowledge acquired from math training is transferable to coding, enabling the model to develop coding abilities subsequently.

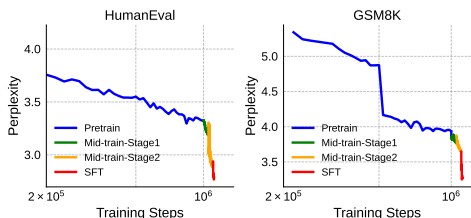

Figure 7: Evolution of reasoning capability during training, measured by perplexity reductions on reasoning-focused benchmarks: HumanEval for coding and GSM8K for math.

In the following, we position our trained `MobileLLM-R1` within the context of prior state-of-the-art models and compare their performance. We present results for two sets of models: the base models, evaluated after pre-training and mid-training, and the final models after the complete training pipeline. The experimental settings and full training pipeline can be found in Section A

**Base Model** Figure 8 compares base reasoning models across multiple benchmarks. We group them into fully open-source models (OLMo OLMo et al. (2024), SmolLM Allal et al. (2025), `MobileLLM-R1`), with weights, data, and training recipes available, and partially open-source models (Qwen Yang et al. (2025), Gemma Team et al. (2025), LLaMA Dubey et al. (2024)), which released model weights and partial training procedures. Compared to fully open-source models, `MobileLLM-R1` consistently outperforms both OLMo and SmolLM across all parameter scales. For example, at the 140M scale, `MobileLLM-R1` achieves 16.3% GSM8K and 15.9% HumanEval, dramatically surpassing SmolLM2-135M (1.8% and 0.0%, respectively). Compared to prior partially open-source models, such as Qwen3-0.6B, `MobileLLM-R1` achieves comparable or superior results despite being trained on substantially fewer tokens (4.2T for `MobileLLM-R1` vs. 36T for Qwen3). Notably, `MobileLLM-R1`-950M attains the highest HumanEval score (46.3%) among all sub-1B models, significantly outperforming Qwen3-0.6B (30.5%).

**Post-trained Model** Figure 9 presents the performance of post-trained models. Notably, on Live-CodeBench, small models below 400M parameters struggle to produce reliable outputs. In contrast, `MobileLLM-R1`-360M achieves 5.1 points, surpassing even models with over 1B parameters, such as SmolLM2-1.7B, Gemma3-1B, and LLaMA3.2-1B. Remarkably, `MobileLLM-R1`-950M demonstrates a substantial accuracy gain over Qwen3-0.6B on LiveCodeBench and even matches the performance of much larger state-of-the-art models, such as DeepSeek-R1-Distill-Qwen-1.5B. Across Math and AIME benchmarks, `MobileLLM-R1` consistently outperforms other fully open-source models and achieves scores comparable to the partially open-source Qwen3 series. See Appendix B.1 for detailed comparisons.

## 5 RELATED WORK

The advent of GPT-3 (Brown et al., 2020) highlighted the transformative potential of large language models (LLMs), spurring research on both proprietary models (e.g., Claude (Anthropic, 2024)) and open-source alternatives (e.g., LLaMA (Touvron et al., 2023a;b; Dubey et al., 2024), Gemma (Team et al., 2024a;b; 2025), Qwen (Team, 2024; Yang et al., 2025)). Scaling laws have been central to

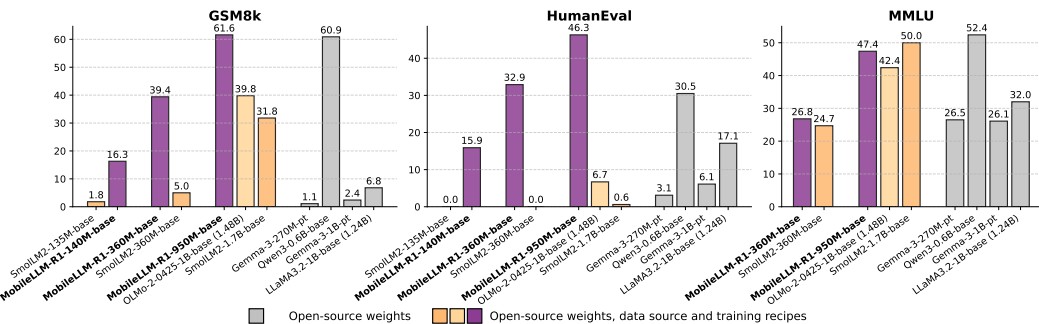

Figure 8: Performance comparison of base models across three tasks: GSM8k, HumanEval, and MMLU. Models are grouped by parameter size and color-coded by model family: `MobileLLM-R1` (purple), SmolLM (orange), OLMo (yellow), and other partially open-source models (gray). Labels indicate model name and size for select models. `MobileLLM-R1` consistently achieves strong performance across tasks while remaining parameter-efficient. A comprehensive comparison is presented in Table 8.

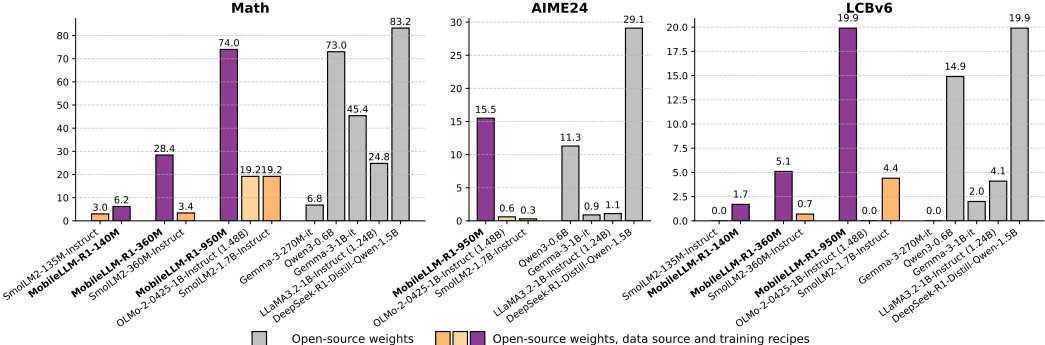

Figure 9: Performance comparison of post-trained models across three tasks: MATH, AIME'24, and LiveCodeBench-v6. The full comparison results are provided in Table 9.

understanding how larger models elicit emergent capabilities and potential performance singularities, while efficiency-accuracy trade-offs in smaller models are increasingly studied to optimize computational resources. MobileLLM Liu et al. (2024) pioneered on-device LLM deployment, followed by high-performance small models such as OLMO OLMo et al. (2024) and SmolLM Allal et al. (2025), reflecting a broader trend toward transparency, including open-sourcing weights, training data, and full pipelines.

Research has also shifted from instinctive response to explicit reasoning thinking, exemplified by OpenAI's O1 (Jaech et al., 2024) and DeepSeek-R1 Guo et al. (2025). Qwen Yang et al. (2025) is another example of a high-quality reasoning model, with smaller variants achieving state-of-the-art benchmark results. Prior studies suggest reasoning emerges only after extremely large-scale pretraining Yang et al. (2025). In contrast, our work shows that a small model can attain strong reasoning abilities using only 4.2T pretraining tokens—comparable to Qwen trained on 36T tokens. We release our full data collection and training pipeline with a detailed rationale for each data selection, ensuring maximal transparency and reproducibility.

# 6  CONCLUSION

We present a data-centric framework to maximize reasoning in small language models under limited parameters and tokens. We introduce benchmark-free, self-evolving data optimization, a principled dataset-level weighting method that leverages cross-domain influences to dynamically tailor the data mixture. This approach enables strong performance on code, math, and knowledge benchmarks without exposing any benchmark data during training or mixture construction. Trained on 4.2T tokens drawn from ~2T curated open-source data, `MobileLLM-R1` achieves state-of-the-art results among small models with a fully open-sourced recipe, and matches Qwen3-0.6B with only 11.7% of its 36T-token training data. Our findings challenge the conventional belief that small reasoning models require massive data, instead underscoring the pivotal role of data quality, token efficiency, and principled data curation.

ETHICS STATEMENT

Our work investigates methods for optimizing the training and deployment of small-scale language models. The models and datasets used in this study are publicly available and widely adopted in the research community. We did not collect new human subject data, nor did we rely on sensitive or proprietary sources. Potential risks primarily concern the general risks associated with language models. While these risks are not specific to our contributions, we acknowledge them and emphasize that our focus is methodological rather than on downstream deployment. We encourage future applications of our techniques to carefully assess ethical considerations in their respective domains.

REPRODUCIBILITY STATEMENT

We follow the ICLR reproducibility guidelines. All datasets used are publicly available, as we revealed in Section A.3. We describe data processing procedures, model architectures, training configurations, and hyperparameters in detail in Sections A. We will release code and trained model checkpoints to support full reproducibility.

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

APPENDIX

## A FINAL RECIPE

To ensure reproducibility and to provide a transparent foundation for future research, we begin by detailing our model architecture and training setup, including the curated data sources and the complete recipe. Building upon fully open-source data, `MobileLLM-R1` establishes a new standard among transparent reasoning models, achieving substantially higher accuracy than prior fully open-source efforts such as OLMo and SmolLM.

### A.1 MODEL ARCHITECTURE

Our model architecture is based on designs from MobileLLM Liu et al. (2024) and LLaMA3.2 Dubey et al. (2024). We adopt the LLaMA3.2 tokenizer with a 128k subword vocabulary. We also incorporate QK-norm to mitigate training instabilities in the self-attention block. Following MobileLLM, we adopt weight sharing between the input and output embeddings to improve parameter efficiency. A complete set of architectural specifications is reported in Table 3.

Table 3: Detailed architecture specifications of `MobileLLM-R1`. "Dim" denotes the embedding dimension and "Hidden Dim" represents the dimension inside the feed-forward network.

| Model | Layer | Head | KV-Head | Dim | Hidden Dim | Params |
|---|---|---|---|---|---|---|
| `MobileLLM-R1`-140M | 15 | 9 | 3 | 576 | 2048 | 140.2M |
| `MobileLLM-R1`-360M | 15 | 16 | 4 | 1024 | 4096 | 359.4M |
| `MobileLLM-R1`-950M | 22 | 24 | 6 | 1536 | 6144 | 949.2M |

### A.2 TRAINING RECIPE

The complete set of training hyperparameters is detailed in Table 4, with stage-specific configurations as follows:

**Pre-training phase**: Models are initialized from scratch and optimized with Adam $(\beta_1, \beta_2, \epsilon) = (0.9, 0.95, 10^{-8})$ and weight decay 0.1. The learning rate employs a 2k-step warmup followed by linear decay to $0.1\times$ the peak value.

**Mid-training phase**: Optimization continues with Adam, where the learning rate decays linearly to zero. Knowledge distillation is applied with Llama-3.1-8B-Instruct model as the teacher, where the student is trained by minimizing the KL divergence between its output logits and the teacher's logits.

**Post-training phase**: Adam is used with zero weight decay. The learning rate warmup ratio is set to 0.03 for general-purpose SFT and 0.1 for reasoning-specific SFT, followed by linear decay to zero.

Table 4: Training setup across different stages.

| Stage | Phase | Tokens / Samples | BS | Seq. Len. | Steps / Epochs | LR | #GPUs | Time |
|---|---|---|---|---|---|---|---|---|
| *Pre-training* | Phase 1 | 2T tokens | 16 | 2k | 500k | 4.00E-03 | 16×8 | 4–5 days |
| | Phase 2 | 2T tokens | 16 | 2k | 500k | 4.00E-03 | 16×8 | 4–5 days |
| *Mid-training* | Phase 1 | 100B tokens | 4 | 4k | 50k | 3.60E-04 | 16×8 | 1–2 days |
| | Phase 2 | 100B tokens | 4 | 4k | 50k | 3.60E-04 | 16×8 | 1–2 days |
| *Post-training* | General SFT | 866K samples | 4 | 4k | 2 epoch | 5.00E-06 | 16×8 | ∼2h |
| | Reasoning SFT | 6.2M samples | 8 | 32k | 4 epoch | 8.00E-05 | 16×8 | ∼2.5 days |

### A.3 DATA

#### A.3.1 PRETRAINING

To balance general language understanding with strong reasoning capabilities, we design a two-stage pre-training curriculum with carefully curated data sources (Table 5). Phase 1 emphasizes broad coverage through large-scale web and educational corpora such as FineWeb-Edu, which provide linguistic and domain diversity. At the same time, we seed the mixture with reasoning-rich corpora, including OpenWebMath, Arxiv, and StackExchange, to expose the model early to mathematical and scientific discourse. In Phase 2, we deliberately shift the weighting toward specialized reasoning datasets—such as FineMath, OpenWebMath, Algebraic Stack, and Facebook

Table 5: Pre-training datasets and mixture ratios for the two-stage curriculum, with 2T tokens sampled per phase.

| Dataset | Rows | Tokens (B) | Phase1 Mix | Phase2 Mix |
|---|---|---|---|---|
| StarCoder | 206,640,114 | 263.8 | 10.66% | 0.52% |
| OpenWebMath | 6,117,786 | 12.6 | 6.93% | 23.33% |
| FineWeb-Edu | 1,279,107,432 | 1300 | 63.75% | 54.83% |
| Wiki | 7,222,303 | 3.7 | 5.03% | 0.14% |
| Arxiv | 1,533,917 | 28 | 6.36% | 1.32% |
| StackExchange | 29,249,120 | 19.6 | 5.03% | 0.86% |
| Algebraic stack | 3,404,331 | 12.6 | 2.25% | 1.26% |
| Nemotron science | 708,920 | 2 | – | 0.03% |
| Nemotron code | 10,108,883 | 16 | – | 0.72% |
| Nemotron math | 22,066,397 | 15 | – | 3.01% |
| Cosmopedia | 31,064,744 | 25 | – | 2.70% |
| Facebook natural reasoning | 1,145,824 | 1.8 | – | 3.18% |
| FineMath | 48,283,984 | 34 | – | 8.01% |
| peS2o | 38,800,000 | 50 | – | 0.08% |
| **Total** | | | **100%** | **100%** |

Natural Reasoning—while reducing the proportion of generic sources. This skewed allocation ensures that the model continues to benefit from general pre-training signals while increasingly focusing on structured reasoning tasks, ultimately aligning the training distribution with our goal of developing a compact yet strong reasoning model. The total token count of the datasets used in per-training is 1.8T. We sample data from each source according to the predefined mixture weights, drawing a total of 2T tokens per phase.

### A.3.2    MID-TRAINING

For mid-training, we construct a mixture that complements pre-training by targeting benchmarks and reasoning-intensive domains (Table 6). The first phase emphasizes coverage of general-purpose datasets (e.g., Dolmino DCLM baseline, FLAN, and peS2o) alongside curated knowledge sources such as Wiki and StackExchange. This configuration is intended to improve the model's broad knowledge, reflected in the performance in general benchmarks like MMLU, where factual recall is critical. In the second phase, we deliberately skew the mixture toward math and coding corpora, particularly Dolmino Math, Nemotron-CC-Math, and Nemotron-Code, while reducing the weight of general-purpose datasets. We also introduce a small but targeted set of benchmark-style datasets (e.g., GSM8K, ARC, OBQA) to align training with downstream evaluation. This two-stage weighting scheme allows the model to first consolidate broad knowledge and language understanding before specializing on reasoning-intensive math and code tasks, which are central to our objective of building compact yet competitive reasoning models. The reported data mix ratios correspond to post sub-sampling. We will also release the code for the sub-sampling procedure to ensure full reproducibility for anyone following our protocol. For each mid-training phase, we sample 100B tokens from the sources following the predefined mixture distribution.

Table 6: Mid-training datasets and mixture ratios, with 100B tokens sampled per phase according to the mix ratio.

| Dataset | Subset | Rows (M) | Phase1 Mix | Phase2 Mix |
|---|---|---|---|---|
| Dolmino | DCLM Baseline | 606 | 37.03% | 6.51% |
| | FLAN | 57.3 | 4.10% | 0.72% |
| | peS2o | 38.8 | 11.41% | 2.01% |
| | Wiki | 6.17 | 2.66% | 0.47% |
| | StackExchange | 2.48 | 2.12% | 2.00% |
| | Math | 21 | 11.63% | 29.10% |
| Nemotron | Nemotron-Pretraining-Code-v1 | 882 | 20.69% | 29.10% |
| | Nemotron-CC-Math-v1 | 144 | 3.45% | 19.40% |
| StarCoder | StarCoder | 206 | 6.90% | 9.70% |
| Benchmark Set | TriviaQA (train) OBQA (train) NaturalQuestions (train) PIQA (train) GSM8K (train) BoolQ (train) ARC-Easy (train) ARC-Challenge (train) | ~0.01 | – | 0.97% |
| **Total** | | | **100.00%** | **100.00%** |

### A.3.3 POST-TRAINING

In the post-training stage, we leverage established post-training datasets. Following standard practice, we first align the model with instructions through general supervised fine-tuning (SFT) and then apply reasoning-specific SFT to extend the context length and promote a long chain-of-thought (CoT) reasoning style. The datasets used in post-training are shown in Table 7

Table 7: Post-training data.

| Stage | Dataset | Rows |
|---|---|---|
| *General* | Tulu3-SFT | 866K |
| *Reasoning* | OpenMathReasoning | 3.2M |
| | OpenScienceReasoning-2 | 802K |
| | OpenCodeReasoning-2 | 2.2M |

## B COMPARISON WITH PREVIOUS METHODS

### B.1 DETAILED ACCURACY COMPARISON

Table 8 demonstrates the performance of sub-billion and near-billion parameter reasoning base models across six widely used benchmarks. The results demonstrate that the proposed `MobileLLM-R1` models consistently outperforms prior open-source models of comparable or larger scale. In the <150M regime, `MobileLLM-R1`-140M delivers substantial gains over SmolLM2-135M, particularly on GSM8K (16.3 vs. 1.8) and HumanEval (15.9 vs. 0.0). Similarly, `MobileLLM-R1`-360M surpasses both Gemma-3-270M-pt and SmolLM2-360M, achieving more than double the accuracy on MATH500 and GSM8K. At the higher end, `MobileLLM-R1`-950M matches or exceeds the performance of larger models: it improves upon Qwen3-0.6B in code generation (HumanEval: 46.3 vs. 30.5) and commonsense reasoning (58.6 vs. 55.3), while remaining competitive on math and programming tasks. These results highlight that carefully curated training data, rather than sheer scale, is the key driver of reasoning performance in lightweight models.

In Table 9, we present a comparison of reasoning-oriented models across diverse benchmarks, including MATH500, GSM8K, AIME'24, AIME'25, and LCBv6, all evaluated in zero-shot settings. The results show that `MobileLLM-R1` models deliver substantial improvements over prior sub-billion parameter instruct-tuned baselines. In the <150M and 150M–400M ranges, `MobileLLM-R1` achieves up to 8× higher accuracy on MATH500 and GSM8K compared to SmolLM2-Instruct and Gemma-3-it variants, while also providing consistent gains on LCBv6. At the higher end, `MobileLLM-R1`-950M reaches an AIME'24 score of 15.5 — a level of reasoning performance previously unseen in open models trained on fully transparent datasets—while maintaining competi-

Table 8: Performance comparison of reasoning base models across multiple benchmarks. Here, CommonSense Avg. denotes an average of 8 tasks in CommonSense Reasoning benchmarks including ARC-easy, ARC-challenge, BoolQ, PIQA, SIQA, HellaSwag, OBQA, and WinoGrand. Models with fewer than 150M parameters do not yield reliable MMLU scores and are therefore denoted as '-'.

| Model | Size | MATH500 (4-shot, em) | GSM8K (8-shot, em) | MBPP (3-shot, pass@1) | HumanEval (0-shot, pass@1) | CommonSense (0-shot, avg acc.) | MMLU (5-shot, acc.) |
|---|---|---|---|---|---|---|---|
| *<150M* | | | | | | | |
| SmolLM2-135M | 135M | 0.4 | 1.8 | 3.8 | 0.0 | **50.7** | – |
| **MobileLLM-R1-140M-base** | 140M | **4.6** | **16.3** | **5.4** | **15.9** | 44.3 | – |
| *150M − 400M* | | | | | | | |
| Gemma-3-270M-pt | 268M | 0.6 | 1.1 | 2.0 | 3.1 | 48.4 | 26.5 |
| SmolLM2-360M | 362M | 1.8 | 5.0 | 19.4 | 0.0 | **56.6** | 24.7 |
| **MobileLLM-R1-360M-base** | 359M | **13.4** | **39.4** | **20.8** | **32.9** | 51.0 | **26.8** |
| *400M − 1B* | | | | | | | |
| Qwen2.5-0.5B | 494M | 14.8 | 41.8 | 29.6 | 28.1 | 52.3 | 47.5 |
| Qwen3-0.6B-Base | 596M | **29.8** | 60.9 | 39.0 | 30.5 | 55.3 | **52.4** |
| **MobileLLM-R1-950M-base** | 949M | 26.8 | **61.6** | 39.2 | **46.3** | **58.6** | 47.4 |
| *>1B* | | | | | | | |
| Gemma-3-1B-pt | 1.00B | 0.6 | 2.4 | 9.4 | 6.1 | 57.3 | 26.1 |
| LLaMA3.2-1B | 1.24B | 1.6 | 6.8 | 26.6 | 17.1 | 58.4 | 32.0 |
| OLMo-2-0425-1B | 1.48B | 5.2 | 39.8 | 7.8 | 6.7 | 61.0 | 42.4 |
| Qwen2.5-1.5B | 1.54B | 31.0 | 68.4 | 44.6 | 36.6 | 58.7 | 61.2 |
| SmolLM2-1.7B | 1.71B | 11.6 | 31.8 | 35.4 | 0.6 | 62.9 | 50.0 |
| Qwen3-1.7B-Base | 2.03B | 38.5 | 76.2 | 56.4 | 47.6 | 60.9 | 62.1 |

Table 9: Evaluation results of different reasoning models across MATH500, GSM8K, AIME'24, AIME'25, and LCBv6 benchmarks. For AIME, we evaluate models across 64 runs and report the average accuracy. For LiveCodeBench, results are reported as the average accuracy across 16 runs. Models with fewer than 400M parameters do not produce reliable AIME scores and are therefore denoted as '-'.

| Name | Size | MATH500 0-shot pass@1 | GSM8K 0-shot pass@1 | AIME'24 0-shot pass@1, n=64 | AIME'25 0-shot pass@1, n=64 | LCBv6 0-shot pass@1, n=16 |
|---|---|---|---|---|---|---|
| *<150M* | | | | | | |
| SmolLM2-135M-Instruct | 135M | 3.0 | 2.4 | – | – | 0.0 |
| **MobileLLM-R1-140M** | 140M | **6.2** | **4.1** | – | – | **1.7** |
| *150M – 400M* | | | | | | |
| Gemma-3-270M-it | 268M | 6.8 | 8.4 | – | – | 0.0 |
| SmolLM2-360M-Instruct | 362M | 3.4 | 8.1 | – | – | 0.7 |
| **MobileLLM-R1-360M** | 359M | **28.4** | **24.5** | – | – | **5.1** |
| *400M – 1B* | | | | | | |
| Qwen2.5-0.5B-Instruct | 494M | 31.2 | 48.1 | 0.1 | 0.3 | 3.6 |
| Qwen3-0.6B | 596M | 73.0 | **79.2** | 11.3 | **17.0** | 14.9 |
| **MobileLLM-R1-950M** | 949M | **74.0** | 67.5 | **15.5** | 16.3 | **19.9** |
| *>1B* | | | | | | |
| Gemma-3-1B-it | 1.00B | 45.4 | 62.9 | 0.9 | 0.0 | 2.0 |
| LLaMA3.2-1B-Instruct | 1.24B | 24.8 | 38.8 | 1.1 | 0.2 | 4.1 |
| OLMo-2-0425-1B-Instruct | 1.48B | 19.2 | 69.7 | 0.6 | 0.1 | 0.0 |
| OpenReasoning-Nemotron-1.5B | 1.54B | 83.4 | 76.7 | 49.7 | 40.4 | 28.3 |
| DeepSeek-R1-Distill-Qwen-1.5B | 1.54B | 83.2 | 77.3 | 29.1 | 23.4 | 19.9 |
| Qwen2.5-1.5B-Instruct | 1.54B | 54.0 | 70.0 | 2.5 | 0.9 | 7.9 |
| SmolLM2-1.7B-Instruct | 1.71B | 19.2 | 41.8 | 0.3 | 0.1 | 4.4 |
| Qwen3-1.7B | 2.03B | 89.4 | 90.3 | 47.0 | 37.0 | 29.8 |

tive accuracy on GSM8K (67.5) and surpassing Qwen3-0.6B in LCBv6 (19.9 vs. 14.9). Although larger proprietary models such as Qwen3-1.7B and OpenReasoning-Nemotron-1.5B achieve stronger absolute results, `MobileLLM-R1` demonstrates that strong reasoning can emerge with far fewer training tokens when coupled with careful data curation and resampling. These findings underscore the central claim of this work: reasoning capabilities in lightweight models are not contingent on scaling to massive proprietary corpora.

## C ON-DEVICE PROFILING

We benchmark the latency of `MobileLLM-R1` models (140M, 360M, 950M) and LLaMA-3 models (1B, 3B, 8B) using ExecuTorch on a Samsung Galaxy S22 (8 GB RAM). In this experiment, we run each model with a warm-up phase and then average the results over three iterations using the same prompt. On-device applications usually adopt quantized settings. To reflect typical on-device deployment, models are benchmarked under quantized settings: linear layers are quantized using 8-bit dynamic activations and 4-bit weights. For the 125M model, a group size of 32 is adopted since its hidden dimension is not divisible by 128, whereas all larger models use a group size of 128. Embedding layers are quantized to 4-bit with a group size of 32.

Latency is reported in tokens per second across generation lengths ranging from 1k to 32k. As expected, latency decreases as context length grows and as model size increases. When context length becomes large, the KV cache size approaches the size of the weights. Since we are already memory-bound when loading weights, the additional cost of loading the KV cache is no longer negligible.

For long-context reasoning, the sequence length can rapidly extend to 32k tokens. At this scale, larger models quickly hit memory constraints. Thus, enabling efficient long-context on-device reasoning necessitates smaller models. Beyond memory efficiency, smaller models also provide significantly higher generation throughput, as speed scales approximately inversely with model size, underscoring the importance of compact models for practical on-device reasoning.

Table 10 reports the measured generation throughput (tokens per second) of `MobileLLM-R1` models and LLaMA-3 baselines across different context lengths on the Samsung Galaxy S22. The results confirm the clear efficiency advantage of smaller models: the 140M variant sustains over 100 tokens/s up to 8k tokens, while even the 360M and 950M models maintain practical throughput across medium-length contexts. In contrast, larger models such as LLaMA-3.2-3B and LLaMA-3.1-8B

Table 10: Generation speed (token/s) across context lengths for different model sizes. OOM indicates out-of-memory.

| Model | Size | 1k | 2k | 4k | 8k | 16k | 32k |
|---|---|---|---|---|---|---|---|
| MobileLLM-R1-140M | 140M | 129.67 | 116.16 | 111.42 | 110.47 | 96.38 | 79.71 |
| MobileLLM-R1-350M | 360M | 77.23 | 61.54 | 61.24 | 61.27 | 61.13 | OOM |
| MobileLLM-R1-900M | 950M | 31.05 | 27.59 | 25.09 | 25.36 | OOM | OOM |
| LLaMA3.2-1B | 1.24B | 28.71 | 24.99 | 21.99 | 22.00 | OOM | OOM |
| LLaMA3.2-3B | 3.21B | 9.04 | 8.61 | OOM | OOM | OOM | OOM |
| LLaMA3.1-8B | 8.03B | 2.76 | OOM | OOM | OOM | OOM | OOM |

encounter out-of-memory failures beyond short contexts, and the LLaMA-3.2-1B model runs at less than one-quarter the speed of MobileLLM-R1-140M. Moreover, long-context settings (16k–32k) quickly exhaust memory capacity for billion-scale models, highlighting the prohibitive cost of KV-cache storage at scale. These findings demonstrate that sub-billion parameter models not only enable reasoning with reduced memory overhead but also achieve far higher generation speed, making them far better suited for on-device long-context inference.

# D  ABLATION ANALYSIS AND INSIGHTS

## D.1  LEARNING RATE AFFECTS REPRESENTATION LEARNING

We conduct an ablation study on the effect of learning rate during pretraining. The setup employs 16 nodes with 8 GPUs each, a per-device batch size of 8, and training for 500k steps with a sequence length of 2048, corresponding to approximately 1T tokens. The learning rate follows a linear decay schedule from its peak value to 10% of the maximum, with an initial warmup phase of 2k steps. The mid-training stage follows the same optimization recipe as used in our final configuration.

Our results indicate that, while pretraining MMLU performance remains largely unchanged across learning rates, the differences become significant after mid-training, as shown in Table 11. Models pretrained with larger learning rates exhibit superior downstream accuracy, suggesting that they acquire stronger intermediate representations during pretraining. Importantly, pretraining accuracy itself does not provide predictive power for downstream outcomes.

Therefore, we investigate structural diagnostics of pretrained representations that can act as low-cost proxies for downstream generalization. We draw inspiration from RankMe Garrido et al. (2023), originally proposed in the context of vision self-supervised finetuning:

$$\text{RankMe}(Z) = \exp\left(-\sum_{k=1}^{\min(N,K)} p_k \log p_k\right), \quad \text{where } p_k = \frac{\sigma_k(Z)}{\sum_{i=1}^{\min(N,K)} \sigma_i(Z)} + \epsilon, \tag{9}$$

where $\sigma_k(Z)$ denotes the singular values of output embeddings $Z$. Our analysis indicates that downstream performance is closely tied to parameter utilization during pretraining: models that activate broader, task-relevant subspaces and maintain high-rank embeddings facilitate fine-tuning to more effectively recover and exploit useful representations.

Table 11: Performance comparison between pre-training and after mid-training under different learning rates.

| | Pre-training | | After Mid-training |
|---|---|---|---|
| Learning Rate | MMLU | RankMe Score | MMLU |
| 4e-3 | 27.92 | 21.98 | 36.31 |
| 2e-3 | 27.67 | 20.19 | 33.95 |
| 1e-3 | 26.87 | 14.22 | 29.02 |
| 4e-4 | 27.50 | 7.81 | 27.55 |

In our case, the RankMe score correlates strongly with post mid-training MMLU accuracy, supporting its utility as a proxy measure that could potentially provide early signals and save resources. Furthermore, analysis of the RankMe score shows that higher learning rates lead to higher representational rank. This implies that a larger fraction of parameters are effectively utilized during pretraining, which in turn enables stronger generalization and improved final task accuracy. We note that this is a preliminary study and represents a promising direction for future investigation.

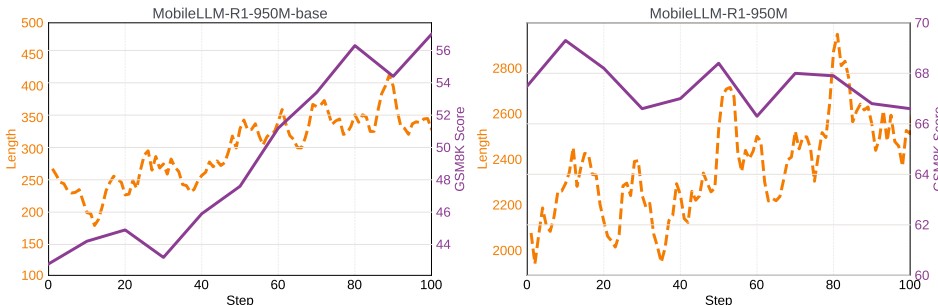

Figure 10: Accuracy and sequence length during RL fine-tuning on `MobileLLM-R1` base model and final model after supervised finetuning(SFT). Small models benefit from RL when pretrained on a suitable corpus, with `MobileLLM-R1`-950M-base showing steady reasoning gains. However, small model trained with SFT data outperforms base+RL, and additional RL can degrade small models optimized with SFT.

## D.2 Discussion: Whether to Use RL or Not

A central question in the development of small reasoning language models is whether reinforcement learning (RL) is beneficial. Recent works such as DeepSeek-R1 (Guo et al., 2025) and Qwen3 (Yang et al., 2025) adopt a two-stage paradigm: (i) train a large model with RL, and (ii) generate strong reasoning traces to distill smaller student models, either via sequence-level distillation or a combination of logit-level and sequence-level distillation. The underlying rationale is that large models, equipped with greater exploration capacity and self-improvement potential, can discover more effective reasoning trajectories, while small models, being capacity-constrained, are better suited to imitating curated demonstrations rather than performing exploration directly.

To assess the impact of reinforcement learning (RL) on small reasoning models, we perform an ablation study by applying an RL stage to both the base model and the final model, i.e., fine-tuned with SFT data. The base model is finetuned for 100 steps on the TULU3 dataset (1% of total data) as a cold start, solely to learn the correct output format. Subsequently, both the base and SFT models undergo GRPO training on the NuminaMath-TIR dataset with a learning rate of $3 \times 10^{-6}$, 100 training steps, a batch size of 32 prompts, and 8 generations per prompt. The KL coefficient for the reference model is set to $\beta = 0$.

The results in Figure 10 highlight several key observations: (1) Small models can also benefit from RL-based fine-tuning when they are well-pretrained on a suitable corpus. Results show that `MobileLLM-R1`-950M-base achieves evident improvements in reasoning accuracy as the average length gradually increases. (2) Supervised fine-tuning (SFT) data distilled from large models, with data source listed in Table 7, consistently yields higher performance than directly applying RL to small models, corroborating prior findings. For instance, the final GSM8K accuracy of RL-optimized `MobileLLM-R1`-950M-base is 57.0, compared to 74.0 for the SFT-trained `MobileLLM-R1`-950M. (3) For high-performing small models fully fine-tuned on SFT data, additional RL does not observe a significant performance improvement. This suggests that SFT provides more structured and reliable supervision than the noisy self-exploration signal accessible to small models, which often lack the capacity to further refine their reasoning policies beyond the distilled demonstrations.

## D.3 Computational Overhead of the Data Curation Pipeline

To provide a detailed characterization of the computational overhead of our method, we measure and report the data curation costs, including representative dataset construction, influence score computation and leave-one-out ablation.

- Hierarchical Rejection Sampling for producing representative datasets: ~200 GPU hours.
- Pre-training influence computation: ~5,100 GPU hours.
- Mid-training influence computation: ~1,100 GPU hours.
- Leave-one–out ablation: ~50 GPU hours per dataset, ~ 400 GPU hours in total.
- Total: ~6,800 GPU hours.

Training cost:

- Pre-training: ∼28,000 GPU hours.
- Mid-training: ∼10,000 GPU hours.
- Post-training: ∼600 GPU hours.
- Total: ∼38600 GPU hours.

The detailed training cost can be found in Table 4. Our analysis indicates that the data curation pipeline itself is also computationally efficient: the combined cost of data curation and ablation experiments constitutes only approximately 15% of the total cost of data curation and model training.

## E    USE OF LLM

We leverage large language models (LLMs) to assist in drafting and polishing written content. Specifically, LLMs are used to improve clarity, coherence, and readability of the text, while ensuring that technical accuracy and intended meaning are preserved. All outputs generated by the LLM are carefully reviewed and edited by the authors to maintain factual correctness and align with the scientific content. The LLM serves as a tool to support writing efficiency, not to generate original research ideas or conclusions.

