# OpenReview forum: "MobileLLM-R1: Exploring the Limits of Sub-Billion Language Model Reasoners with Open Training Recipes"
_ICLR.cc/2026/Conference — ICLR 2026 Poster_

### Official Review · Reviewer_9Hby · 2025-10-31

**Soundness:** 3
**Presentation:** 3
**Contribution:** 3
**Rating:** 6
**Confidence:** 4

**Summary:**

This paper challenges the prevailing assumption that achieving strong reasoning capabilities necessitates massive training datasets (e.g., >10T tokens). The authors propose a data-centric and highly curated training pipeline, introducing a series of sub-billion parameter language models called X-LLM-R1 (140M, 360M, 950M).

The core contribution of this work is a fully open-sourced "recipe" for efficiently training small reasoning models, which features two main stages of innovation:
1.  **Pre-training Stage:** A "Datamixing via Cross-Capability Self-Influence" strategy is proposed. This benchmark-free method automatically optimizes the mixing ratios of various open-source datasets by calculating the influence scores of data samples on "capability-probing datasets" (covering code, math, and knowledge).
2.  **Mid-training Stage:** A "data-model co-evolution" strategy for knowledge compression is introduced. This strategy dynamically computes sample influence scores during training and iteratively removes (filters out) samples with zero or negative influence, enabling the model to absorb knowledge more efficiently.

Experimental results show that the X-LLM-R1-950M model, trained on only ~2T tokens of curated open-source data (4.2T tokens total pre-training), matches or even surpasses the performance of Qwen3-0.6B (trained on 36T tokens) on several reasoning benchmarks, particularly AIME and HumanEval. The authors commit to releasing all data sources, mixing ratios, models, and code.

**Strengths:**

1.  **Important Research Problem:** The paper tackles a significant research question: how to train small, efficient, and deployable reasoning models. Its argument for "quality over quantity" in data provides a valuable path for researchers with limited resources.
2.  **Novel Methodology:** The core contributions (influence-score-based pre-training mixture and mid-training knowledge compression) are novel and principled. The "benchmark-free" nature of these methods is a key strength, avoiding overfitting to downstream benchmarks.
3.  **Strong Empirical Results:** The results are highly competitive. X-LLM-R1 significantly outperforms other fully open-source models (like OLMo, SmolLM) at all parameter scales. The fact that X-LLM-R1-950M (4.2T tokens) achieves performance comparable to or better than Qwen3-0.6B (36T tokens) on key reasoning benchmarks is a strong demonstration of the recipe's effectiveness.
4.  **Commendable Transparency and Reproducibility:** The paper excels in its commitment to openness. The authors promise to provide full details of their training recipe, including all data sources, mixing ratios, architecture, and hyperparameters. This transparency is of great value to the community.
5.  **Insightful Ablation Studies:** The paper provides valuable insights through detailed ablations, such as the LOO analysis of data sources, the effect of learning rates, and the SFT vs. RL discussion.

**Weaknesses:**

1.  **Implicit Computational Cost of Curation:** The paper emphasizes its "token efficiency" but does not explicitly discuss the computational cost of the data curation process itself. For instance, (1) the LOO analysis requires training multiple models; (2) calculating influence scores, while scalable, also requires significant compute (e.g., training domain-specific checkpoints). A comparison of the *total compute* (curation compute + training compute) versus a "brute force" approach (like Qwen3's 36T) would make the "efficiency" argument more complete.
2.  **Sensitivity to "Capability-Probing Datasets":** The pre-training data mix relies heavily on the constructed "capability-probing datasets." Although this process is "benchmark-free," the construction itself (e.g., using Ask-LLM with specific prompts for hierarchical rejection sampling) introduces designer priors. The paper lacks a sensitivity analysis on how robust the final data recipe is to changes in the design of these probing datasets.
3.  **Trade-off between Reasoning and Knowledge:** The results in Table 8 show that while X-LLM-R1-950M-base excels on GSM8K and HumanEval, it lags behind Qwen3-0.6B-Base and SmolLM2-1.7B-base on MMLU. This suggests that the curation strategy, which is highly optimized for code and math reasoning, might come at the cost of retaining broad factual knowledge.

**Questions:**

1.  **Regarding Curation Compute Cost:** Could the authors provide an estimate of the computational overhead required for the full data curation process (LOO analysis, influence score calculation, etc.)? How does the *total compute budget* (curation + training) of this method compare to the budget needed to simply train a baseline (like Qwen3-0.6B) on 36T tokens?
2.  **Regarding Sensitivity to Probing Datasets:** The pre-training mix depends on the "capability-probing datasets." How much would the final data recipe and model performance change if the construction of these datasets were altered (e.g., different Ask-LLM prompts, or a different sampling threshold than 10%)?
3.  **Regarding the Knowledge vs. Reasoning Trade-off:** The MMLU performance in Table 8 suggests a potential trade-off where optimizing for reasoning (especially in pre-training) may reduce general knowledge. Was this an intentional design choice, or do the authors view this as an unavoidable trade-off for small models with limited capacity?
4.  **Regarding Mid-training Convergence:** In Section 3, the model undergoes two "knowledge compression" stages. Why were two stages chosen? As shown in Figure 5, the influence scores are indeed compressed in Stage 2, but have they fully converged (i.e., reached the "zero or negative influence" state)? Could the authors provide the percentage of data that was filtered out in each stage?

---

> ### Author Response · Authors · 2025-11-21
>
> Thank you for the positive assessment of our paper. We respond to each of your comments and questions as follows:
>
> [ **Computational cost of data curation** ]
>
> Sure, the influence score calculation and data sampling cost is measured and listed as follows:
>
> - Hierarchical Rejection Sampling for producing representative datasets: ~200 GPU hours.
>
> - Pre-training influence computation:  ~5,100 GPU hours
>
> - Mid-training influence computation:  ~1,100 GPU hours.
>
> - Leave-one–out ablation: \~50 GPU hours per dataset, \~400 GPU hours in total. We only train these LOO models for 50k iterations on 2 GPU nodes (\~26B tokens).
>
> - Total: ~6,800 GPU hours
>
> Training cost:
>
> - Pre-training: ~28,000 GPU hours
>
> - Mid-training: ~10,000 GPU hours
>
> - Post-training: ~600 GPU hours
>
> - Total: ~38600 GPU hours
>
> The detailed training cost can be found in Table 4 in the paper.
>
> Therefore, the total cost of data curation and ablations accounts for only about 15% of the overall cost of data curation plus model training. Thanks for the insightful suggestion. We have included these numbers in the paper to provide a clearer computational cost breakdown of our method.
>
> As for the curation–training cost comparison with Qwen, an apple-to-apple comparison is difficult because Qwen does not transparently disclose the entire data curation pipeline or its associated computational cost. Qwen3 is also not a “brute-force” approach; it clearly relies on sophisticated curation techniques. Given that our curation cost is only about 15% of our training cost, our pipeline still offers a clear computational advantage relative to Qwen’s 36T-token training setup (approximately 300,000 GPU-hours), even without accounting for their curation overhead.
>
>
> [ **Sensitivity to Capability-Probing Datasets** ]
>
> Good question. In response, we conducted a sensitivity analysis by varying key design choices in the capability-probing dataset. Specifically, we considered different Ask-LLM prompts and altered the sampling threshold from the default 10%, resulting in different probing dataset sizes: we evaluated using probing dataset sizes of 1K, 5K, and 10K examples.
>
> As summarized in the table below, the inferred sampling ratios remain largely consistent across settings. Notably, fineweb-edu consistently dominates the mixture, while coding, math, and wiki datasets maintain roughly ~5% of the total ratio. It is worth noting that for datasets with small proportions in the mixing ratio, the absolute differences should be considered, since the weighting is computed over the entire dataset. Given the short rebuttal timeframe, we performed only a single sampling run and were unable to complete the pretraining runs. We will report variance across multiple runs and also evaluate accuracy variance in the final version.
>
>  | Dataset | 10K | 5K | 1K |
>  | - | - | - | - |
>  | StarCoder | 10.7% | 11.8% | 6.8% |
>  | OpenWebMath | 6.9% | 4.5% | 8.0% |
>  | FineWeb-Edu | 63.8% | 65.9% | 64.6% |
>  | Wiki | 5.0% | 3.9% | 8.5% |
>  | Arxiv | 6.4% | 7.5% | 7.3% |
>  | StackExchange | 5.0% | 1.0% | 2.7% |
>  | AlgebraicStack | 2.3% | 5.3% | 2.1% |
>
>
> We further evaluated the effect of changing the prompt. The original prompt was:
>
> ```
> ###
> {text}
> ###
> Does the previous paragraph demarcated within ### and ###
> contain informative signal for pre-training a large-language model?
> An informative datapoint should be well-formatted, contain some
> usable knowledge of the world, and strictly NOT have any harmful,
> racist, sexist, etc. content.
>
> {append_value}
> ```
>
> and the new prompt was:
>
> ```
> ###
> {text}
> ###
> Can the text found between the ### markers be considered a valuable source of information for the pre-training process of a large language model? When evaluating whether a data point is informative, it is important to ensure that the content is presented in a clear and organized manner.
>
> {append_value}
> ```
>
> Despite the substantial differences between the two prompts, the overall trend in the final derived datamix remains consistent, with the absolute deviations less than 5%.
>
> | Source | Original Prompt | New Prompt | Absolute Deviation |
> |-|-|-|-|
> | StarCoder | 10.7% | 14.4% | 3.8% |
> | OpenWebMath | 6.9% | 2.7% | -4.3% |
> | FineWeb-Edu | 63.8% | 59.7% | -4.0% |
> | Wiki | 5.0% | 2.3% | -2.7% |
> | Arxiv | 6.4% | 8.8% | 2.5% |
> | StackExchange | 5.0% | 6.0% | 1.0% |
> | AlgebraicStack | 2.3% | 6.0% | 3.8% |
>
> We included these tables and discussions in the revision. This variance analysis may help the audience better understand the robustness of our method.

---

> ### Author Response · Authors · 2025-11-21
>
> [ **Trade-off between reasoning and knowledge in small language models** ]
>
> Thank you for raising this important point. Balancing reasoning and factual knowledge is indeed a key challenge for small models with limited capacity. Different capabilities interact in different ways. Some domains, such as math and coding, are highly complementary—both benefit from improvements in deductive reasoning and structured problem solving. We observe that curated math and code data can mutually reinforce each other. Knowledge and reasoning are not entirely separate either. For instance, solving contextual math word problems requires basic physical or commonsense knowledge (e.g., understanding that distance = speed × time).
>
> We did not deliberately optimize reasoning at the cost of knowledge. However, the availability and quality of open-source datasets vary across domains. Currently, high-quality math and coding corpora are more mature and accessible than broad factual knowledge datasets tailored to MMLU-style evaluation. On the other hand, knowledge-intensive benchmarks pose inherent challenges for small language models, as their limited parameter capacity constrains both the amount of information they can store. Even so, the performance gap is relatively small compared to Qwen3 trained on proprietary data (47.4 vs. Qwen3's 52.4), and we believe that better open-source knowledge datasets would further reduce this difference. Additionally, models like SmolLM2-1.7B benefit from a much larger parameter budget (nearly 2 times), making the trade-off across domains inherently easier to manage.
>
> [ **Mid-training Convergence; Why were two stages chosen?** ]
>
> Great question! The mid-training process only comprises two stages because we filter out samples with zero or negative influence scores. By the end of stage two, very few samples have a positive influence on the model. As a result, the process naturally terminates.
>
> Looking at the histograms in Figure 8, we can see that in stage two, there is no density in the positive influence region; most densities are clustered around zero, with some extending into the negative range. We interpret this distribution as a signal that the sampling-retraining loop has effectively converged.
>
>
> [ **The percentage of data that was filtered out in each stage** ]
>
> Of course, the following table shows data retained after two filtering phases. Phase 1 removes roughly 60–70% of most sources. In Phase 2, most data quickly acquire non-positive influence scores, and the proportion of usable data in most sources is reduced to under 10%.
>
> | Source | phase 1 | phase 2 |
> |-|-|-|
> | DCLM | 30.87% | 2.43% |
> | Math | 26.59% | 1.63% |
> | FLAN | 28.03% | 6.06% |
> | Wiki | 26.11% | 2.78% |
> | peS2o | 29.92% | 5.88% |
> | StackExchange | 30.18% | 2.07% |
>
> We included these results in the appendix to provide a more complete and thorough presentation.
>
>
> *If our responses have satisfactorily addressed your questions, we would be grateful if you could kindly consider further raising the score! We sincerely appreciate your valuable feedback for our submission.*

---

### Official Review · Reviewer_dhHj · 2025-10-31

**Soundness:** 4
**Presentation:** 4
**Contribution:** 4
**Rating:** 8
**Confidence:** 4

**Summary:**

In this paper the authors explore whether it’s possible to obtain strong reasoning capacity in sub-billion parameter models, without using proprietary data or an enormous compute budget. They introduce the X-LLM-R1 family of models ranging from 140M to 950M parameters, trained only on ~ 4.2T tokens. (NB: compare to Qwen-3, this is ~ 12% of the data). The achieve state of art reasoning results among open models, and demonstrate this ability comes more from data quality than pure scale, and show how to co-evolve the model and data using an influence score to focus the training procedure on data that will still contribute to the learning.

The authors also will (/ have but can’t link due to double blind review) release the data, training recipe, architecture implementation, making this an extremely transparent piece of work.

Overall it was a pleasure to read this paper, and I hope my comments provide a useful perspective.

**Strengths:**

- This paper presents an extremely clear example of the separation of pre-, mid-, post-training and fine-tuning - and gives strong evidence of the importance of each. This makes for an extremely valuable resource for others to build upon.

    - As does the open release of the model.

- The performance on key benchmarks is extremely good - especially for an open source, sub-billion parameter model.

- The LLO analysis offers valuable insight into the data mixtures required during an efficient pre-training phase, and which of these are the most influential.

- The key novelty presented in the paper is the use of an influence sampling / weighting method that computes the impact of training data on a approximate sample wise basis on the “probing datasets” - this allows the training corpus to be resampled between each phase during mid training to focus on high utility datasets, making the training process extremely efficient.

- The removal of low / negatively impacting data samples offers a clear and seemingly extremely efficient way to train models using significantly fewer tokens than naively would be expected.

**Weaknesses:**

- The focus of the paper is on the reasoning ability of the model - and makes only very passing comment (lines 362/379) on the impact of the knowledge retention / general usability of the model. Could further comment be provided on how the language understanding, factual retention, and general instruction alignment is affected with such a strong emphasis on reasoning during training?

- The method as presented is powerful - however the authors only show the influence sampling at two points during training (I believe) - between the two pre-training phases and between the two mid training phases. Could comment be provided on whether the authors think additional weighting phases would be more impactful? Or how frequently one could plausibly use such a method? To only compute the influence and weight twice might not be so different to hand curated weightings of data mixtures based on intuition. And that the real power of such a method might be in the ability to compute this frequently?

- There is obviously a compute cost associated with this sampling / probing approach - could the authors comment on this and put it in the context of the relatively small token budget required? One assumes the cost of this influence approach is not excessive, but it is noted the influence is computed only on a subset of the data.

- Additionally the weighting is applied on a dataset by dataset basis - what scope is there for differentiating between samples in a dataset which are well understood by the model vs individual samples which are poorly described?

**Questions:**

1. Would it be possible to add a set of evaluations on the models general competence compared to Qwen (ideally others, but specifically as the closet similar model) to show if there is a trade off for general ability vs reasoning specific ability? (e.g. HeelaSwag, PIQA, NaturalQuestions… etc)

2. The leave one out analysis is really nice, but I was curious about the impact on downstream benchmarks rather than just the NLL? Would it be possible to see the impact of leaving out specific datasets on the reasoning tasks rather than just the model NLL? Alternatively could the authors comment on the effect / importance of the LLO beyond just the increase in pre-training loss and what this says about pre-training data mixtures?

3. You reweight the data at two boundaries. Do you expect more frequent reweighting to help? And where would the diminishing returns point be?

4. What is the total compute overhead of training a model in this way? (In GPU days) Relative to one epoch of normal training? i.e. can you show the cost of running the evaluations for the influence as part of the analysis?

5. I would also really appreciate to see how much of the dataset is required to be sampled when computing the influence to compute the weights?

6. Can you comment on / provide the impact of weighting on each of the datasets? ie. how much do different datasets get weighted up / down at different times, and how predictable is this? Do you need to compute the influence for new models, or is this relative proportion more or less static? In which case would it be possible to take the proportions and train a similar model without the influence step?

---

> ### Author Response · Authors · 2025-11-21
>
> Thank you for your recognition of our work! Below, we provide point-by-point responses to each of your questions.
>
>
> [ **How are the language understanding, factual retention, and general instruction alignmen affected with strong emphasis on reasoning?** ]
>
> That’s an insightful question. We evaluate the X-LLM-950M-base and X-LLM-950M models, before and after fine-tuning on extensive reasoning-focused SFT data, and compare them with llama-1B and llama-1B-instruct, which serve as general-purpose baselines not specifically optimized for reasoning. The table below reveals a pattern that broadly aligns with expectations.
>
> | Model | RACE | TruthfulQA (mc1/mc2) | IFeval (inst_level_loose / inst_level_strict) |
> |-|-|-|-|
> | X-LLM-950M-base | 0.377 | 0.2754 / 0.4445 | 0.4245 / 0.3897 |
> | X-LLM-950M | 0.3464 | 0.299 / 0.461 | 0.2518 / 0.2302 |
> | llama-1B | 0.3789 | 0.2326 / 0.3766 | 0.1906 / 0.1835 |
> | llama-1B-instruct | 0.378 | 0.2717 / 0.4389 | 0.5540 / 0.5036 |
>
> We use RACE [1], TruthfulQA [2], and IFeval [3] to assess language understanding, factual calibration, and general instruction alignment, respectively. For X-LLM, we observe some improvement on TruthfulQA after reasoning-oriented SFT, suggesting that reasoning-oriented data may enhance factual grounding or reduce hallucinations. However, the same fine-tuning leads to decreased performance on RACE and a substantial degradation on IFeval. This may be due to the model’s limited capacity being redistributed—pulled toward reasoning-style templates and away from broad reading comprehension and general instruction-following—or because there is not enough instruction-alignment data in the X-LLM SFT stage, causing it to drift away from strict instruction following. Generally speaking, the reasoning-heavy SFT induces a specialization trade-off.
>
> In contrast, llama-1B, which is tuned for general-purpose behavior, shows the opposite pattern. The llama-1B-instruct model exhibits a large gain on IFeval relative to llama-1B, demonstrating that generic instruction tuning reliably improves alignment behaviors. However, this comes with only modest improvements on TruthfulQA and almost no measurable gain on RACE. Moreover, llama-1B exhibits substantially lower performance than X-LLM-R1-950M on a range of reasoning-intensive benchmarks, including MATH500, GSM8K, AIME’24, AIME’25, and LCBv6.
>
> Taken together, these results highlight that, at the 1B scale, models do have limited representational bandwidth. Steering them toward one capability—reasoning, alignment, or factuality—tends to reduce capacity allocated to others. In the meantime, it is also possible that incorporating broader instruction-following and alignment data into the X-LLM SFT stage could mitigate some of these trade-offs. We have included these discussions in the paper.
>
> [1] RACE: Large-scale ReAding Comprehension Dataset From Examinations.
> [2] TruthfulQA: Measuring How Models Mimic Human Falsehoods.
> [3] Instruction-Following Evaluation for Large Language Models.

---

> ### Author Response · Authors · 2025-11-21
>
> [ **How frequent should we reweight the data? Does more frequent reweighting help?** ]
>
> This question touches the core of the algorithm. However, reweighting datasets is not “the more frequent, the better.”
>
> Basically after each dataset reweighting, a loss spike occurs due to differences in training data, even with the continuous learning rate. In pretraining, transitioning between phases requires hundreds of billions of tokens for the model to recover disrupted representation, as observed in the loss curves from the original two-phase training. Our preliminary experiments show that adding more phases after 4T tokens, rather than splitting the 4T-token budget into more phases, helps, but yields diminishing returns compared with the first two phases. Given that one main goal of our work is token-efficient training, we limit pretraining to two phases.
>
> In mid-training the number of phase choices in mid-training are determined by both model convergence and data convergence. Model convergence is defined as the point where the training loss plateaus, and we empirically observe that that convergence is achieved at approximately 100B tokens in mid-training. Data convergence is defined as the point at which no remaining samples have a positive influence score, indicating that the information content of the dataset has been fully exploited. We found that after two phases, almost all the data in the dataset has negative or near zero influence score, as shown in Figure 8. Therefore, adding another phase would not provide additional benefits.
>
> Although we adopt a two-phase training recipe for X-LLM-R1 in the paper, this choice reflects a trade-off among the benefits of data reweighting, the model’s adaptation rate, and the training budget. We have included these in-depth discussions in the paper and hope these analyses can provide guidance for future practitioners in determining the optimal number of phases for their specific scenarios, taking into account dataset size, token budget, model scale, and target task characteristics.
>
> [ **Advantages of algorithmic over intuition-based data reweighting** ]
>
> Data-driven reweighting offers several advantages over human-designed heuristics. (1) Heuristic-based approaches typically require iterative training to validate a manually chosen reweighting scheme, which incurs substantial computational overhead. (2) The optimal dataset mixture is not always intuitive and may contradict conventional expectations. (3) Human curation also struggles to balance multiple optimization objectives simultaneously. For example, in our case, the data-driven approach can balance the performance on math, code and knowledge domains and found that increasing the proportion of math data improves both math and coding performance more effectively than increasing the proportion of coding data. Such cross-domain effects vary by task and rely on nontrivial expertise in cross-domain knowledge. Trade-offs are often difficult for humans to anticipate but can be automatically captured through data-driven reweighting.
>
> [ **The computational cost of data curation** ]
>
> The influence score calculation and data sampling cost is measured and listed as follows:
>
> - Hierarchical Rejection Sampling for producing representative datasets: ~200 GPU hours.
>
> - Pre-training influence computation:  ~5,100 GPU hours
>
> - Mid-training influence computation:  ~1,100 GPU hours.
>
> - Leave-one–out ablation: \~50 GPU hours per dataset, \~400 GPU hours in total.
>
> - Total: ~6,800 GPU hours
>
> Training cost:
>
> - Pre-training: ~28,000 GPU hours
>
> - Mid-training: ~10,000 GPU hours
>
> - Post-training: ~600 GPU hours
>
> - Total: ~38600 GPU hours
>
> The detailed training cost can be found in Table 4 in the paper.
>
> Therefore, the total cost of data curation and ablations accounts for only about 15% of the overall cost of data curation plus model training. We have included these numbers in the revision to provide a clearer computational cost breakdown of our method.
>
> [ **Within-dataset sample differentiation** ]
>
> Great question. During mid-training, we do consider the differentiating between samples by computing the influence score for each data sample and performing rejection sampling to remove samples with non-positive influence scores. We subsequently aggregate the individual scores to determine dataset-level weights, as mentioned in lines 306–315 in the manuscript. The mid-training data curation thus consists of two steps: (1) sample-level influence for rejection sampling, and (2) dataset-level influence for adaptive data mixing. While this approach could be extended to pretraining, the computational cost is prohibitive; therefore, we apply only dataset-level reweighting in pretraining, and use both sample-level rejection and dataset-level reweighting during mid-training.

---

> ### Author Response · Authors · 2025-11-21
>
> [ **Evaluations on the models general competence compared to Qwen.  (e.g. HellaSwag, PIQA, NaturalQuestions… etc.)** ]
>
> Sure, we evaluated X-LLM-R1 on HellaSwag, PIQA, NaturalQuestions as well as other commonsense reasoning tasks to further evaluate the models general competence and compare it with Qwen3 0.6B, the SoTA small reasoning language model. The results are shown in the Table below:
>
> | Model | ARC easy | ARC challenge | BoolQ | PIQA | SIQA | HellaSwag | OBQA | WinoGrande | Natural Questions |
> |-|-|-|-|-|-|-|-|-|-|
> | Qwen3-0.6B-Base | 66.2 | 40.3 | 69.4 | 69.7 | 45.5 | 52.9 | **40.7** | 57.9 | 3.7 |
> | X-LLM-R1-950M-Base | **66.3** | **45.3** | **79.6** | **74.2** | **48.4** | **57.2** | 39.8 | **58.2** | **5.7 |
>
> Despite not being explicitly optimized for these tasks, X-LLM-R1 surprisingly outperforms the Qwen3 model on 8 out of 9 benchmarks. We hypothesize that HellaSwag and PIQA probe commonsense reasoning, although not chain-of-thought reasoning, and that X-LLM-R1’s underlying reasoning capability transfers effectively to these tasks. Natural Questions, in contrast, targets factual QA. Strong performance here suggests that X-LLM-R1 internalizes useful world knowledge that generalizes beyond individual benchmarks. We attribute this cross-domain transfer to the benchmark-agnostic data curation pipeline used in training X-LLM-R1.
>
>
>
> [ **The impact of leaving out specific datasets on reasoning tasks** ]
>
> Great point. Following your suggestion, we further evaluate models trained under a leave-one-dataset-out setup on the corresponding benchmarks, as shown in the table below:
>
> | | CommonSense | Math | GSM8K | HumanEval |
> |-|-|-|-|-|
> | w/o StarCoder | 47.66 | 0.68 | 2.96 | 1.22 |
> | w/o OpenWebMath | 47.32 | 0.34 | 1.97 | 2.44 |
> | w/o FineWeb-Edu | 39.64 | 0.10 | 2.05 | 1.83 |
> | w/o Wiki | 46.95 | 0.32 | 1.97 | 1.83 |
> | w/o Arxiv | 47.32 | 0.34 | 1.97 | 2.44 |
> | w/o Algebraic stack | 47.16 | 0.38 | 1.97 | 1.83 |
> | Baseline (with all datasets) | 48.57 | 0.74 | 3.51 | 2.88 |
>
> All models are trained under controlled conditions for 50k iterations using 2 GPU nodes (~26B tokens), and we analyze the accuracy differences as an early indicator of each dataset’s contribution to the target domains. The results align with our NLL analysis: removing fineweb-edu causes performance drops across all benchmarks, from commonsense reasoning to math and code, while domain-specific datasets primarily benefit their respective domains. For example, removing the code dataset (StarCoder) most strongly impacts HumanEval, whereas removing math datasets (OpenWebMath, AlgebraicStack) reduces performance on Math and GSM8K. These findings are now discussed in the revision to provide a more comprehensive view of dataset contributions.
>
>
> [ **What LOO says about pre-training data mixtures** ]
>
> Leave-one-out (LOO) serves to provide a first-pass at datasets to see if it makes sense to include a dataset in pretraining from the statistical point of view. However, it only provides directional signals (i.e. dataset A helps benchmark/domain X) but doesn’t go beyond these signals in the form of the extent of how A helps X, thus not helpful in deriving the actual data mix ratio. We therefore use the influence score based method described in Section 2.2 to determine the weighting of each dataset during pretraining.
>
> [ **The sampled dataset scale** ]
>
> Thanks for the question. The final sampled representative dataset contains 10k samples for each dataset. Since we have 14 datasets to weight in the pretraining corpus, we sample 140k examples in total. As we mentioned in Line 133-134 in the manuscript: “Finally, we perform semantic deduplication across corpora, shrinking each dataset in Table 5 to a subset of roughly 10,000 examples.”

---

> ### Author Response · Authors · 2025-11-21
>
> [ **Heuristic explanations for the dataset weightings at different phases** ]
>
> We report the final data-mix ratios in Tables 5 and 6. A clear trend emerges in the later stages of mid-training is that the algorithm increases the allocation to math data while reducing the proportion of web-crawled data. This behavior is consistent with our intuition: web-crawled data establishes the broad linguistic prior, whereas math data is more influential in shaping the distribution toward specialized mathematical reasoning in the later stages.
>
> Besides, the method also goes beyond heuristics-based reweighting. It automatically balances competing skill objectives, identifies effective allocation strategies under multi-objective trade-offs, and discovers non-obvious reweighting patterns that would be difficult to determine manually, as discussed in our response on the [Advantages of algorithmic over intuition-based data reweighting].
>
>
> [ **Whether it needs to compute the influence score for new models** ]
>
> The data mix ratio does not differ significantly across models. In our initial experiments, we let each of the X-LLM-R1 models (140M, 360M, and 950M) to calculating its own weighting and perform one-phase training. We observed that this approach does not yield a meaningful improvement compared to using a single model to compute the influence scores and the data mix ratio for all sizes.
>
> To provide concrete evidence, we conducted the phase-2 mid-training data reweighting using the 360M model trained in phase 1, instead of the original 950M model in phase 1. The resulting sampling ratios are shown in the figure below.
>
> | Source | original (950M) | new (360M) |
> |-|-|-|
> | DCLM | 6.51% | 3.85% |
> | FLAN | 0.72% | 1.76% |
> | peS2o | 2.01% | 4.54% |
> | Wiki | 0.47% | 3.34% |
> | StackExchange | 2.00% | 5.29% |
> | Math | 29.10% | 26.44% |
> | Nemotron-Pretraining-Code-v1 | 29.10% | 25.25% |
> | Nemotron-CC-Math-v1 | 19.40% | 22.09% |
> | StarCoder | 9.70% | 6.43% |
>
> The absolute deviation is within 5%. It is worth noting that for datasets with small proportions in the mixing ratio, the absolute differences should be considered, since the weighting is computed over the entire dataset. We further used this newly sampled data to train the 360M model. The final accuracy deviates from the baseline by less than 1% and shows no meaningful improvement over the original result.
>
> Based on this, we recommend when the target model size range is not extremely large (e.g., within a 10× parameter scale difference), one can simply compute the data mix ratio using one model size and apply the same ratio across all models, which empirically yields comparable performance as calculating the data mix ratio separately for each model.
>
>
> *Thank you for the thoughtful and in-depth questions. Your comments provide valuable perspectives that help strengthen the technical completeness of our paper. We appreciate the opportunity to clarify these points and to provide additional context for the broader community following the openreview discussion.*

---

### Official Review · Reviewer_3Hy6 · 2025-11-01

**Soundness:** 3
**Presentation:** 4
**Contribution:** 4
**Rating:** 8
**Confidence:** 3

**Summary:**

This paper aims to enhance the reasoning capability of sub-billion-parameter language models through data-centric training strategies.
Technically, the paper presents two key contributions:
Influence-based DataMix for pretraining, which adaptively re-weights training datasets based on their estimated contribution to different reasoning domains.

Mid-Training Knowledge Compression, a dynamic filtering mechanism that removes samples with non-positive influence during mid-training, enabling co-evolution between model and data to reduce redundancy and stabilize learning.
Empirically, the proposed 950M-parameter model outperforms OLMo-2-1.48B and SmolLM-2-1.7B, while matching or exceeding Qwen3-0.6B, using significantly less data and compute. All training recipes, datasets, and checkpoints are reported to be fully open-sourced.

**Strengths:**

* Clear motivation and practical significance: The paper challenges the common “scaling is all you need” assumption and demonstrates that strong reasoning capability can emerge in small models through carefully optimized data recipes.
* Technical contribution is sound: Extends AutoMixer from dataset-level weighting to multi-capability, sample-level influence estimation. The proposed method leverages internal capability-probing datasets to guide data weighting without relying on external benchmarks.
* Experimental validation is concrete: Provides impressive empirical results and demonstrates strong efficiency, achieving competitive or superior performance to larger models with substantially less data and compute.

**Weaknesses:**

* Approximation reliability: The Hessian–vector product (HVP) approximation for influence estimation may introduce noise. Validation is indirect, without small-scale ground-truth comparisons or formal variance analysis.
* Capability definition: Evaluations are limited to three domains—Code, Math, and Knowledge—oversimplifying the diversity of reasoning skills such as planning, commonsense, and multilingual reasoning.
* Some missing baselines: The study lacks comparisons with strong alternatives, including AutoMixer, In-Run Data Shapley.

Chang, Ernie, et al. "Automixer: Checkpoint artifacts as automatic data mixers." arXiv preprint arXiv:2506.21910 (2025).

Wang, Jiachen T., et al. "Data shapley in one training run." arXiv preprint arXiv:2406.11011 (2024).

**Questions:**

See weakness

---

> ### Author Response · Authors · 2025-11-21
>
> Thank you for your recognition of our work! In the following, we address each of your questions individually.
>
> [ **Hessian–vector product (HVP) approximation and variance analysis** ]
>
> In this work, we seek a technique that enables influence-score computation at LLM pretraining scale. Hessian–vector product (HVP) approximation provides such a mechanism by substantially reducing the computational burden. Without HVP, exact influence estimation requires $O(p^3)$ time and $O(p^2)$ memory, which is prohibitive for large models with parameter count $p$. HVP reduces the time complexity to $O(kp)$ and the memory complexity to $O(p)$, making large-scale experiments tractable while avoiding the naive simplification of using only gradient cosine similarity on the validation or probing set. Prior work [1,2] has shown that this approximation yields faithful influence estimates, and we observe similarly reliable behavior in our experiments.
>
> One natural concern is that the probing dataset is a subsampled proxy for the full pretraining corpus. Since the approximation relies on gradient calculation defined over this proxy, the representativeness of the probing dataset and the stability of the resulting data-mixing ratios under different sampling choices becomes an important question. To address this, we conducted a variance analysis by varying the probing dataset size from 1k to 5k to 10k examples. As summarized in the table below, the inferred sampling ratios generally remain consistent across settings: fineweb-edu consistently dominates the mixture, and coding, math, and wiki datasets all remain at approximately ~5% of the total ratio. The absolute deviation is less than 5%. It is worth noting that for datasets with small proportions in the mixing ratio, the absolute differences should be considered, since the weighting is computed over the entire dataset. This variance analysis may help the audience better understand the robustness of our method, and we have included these tables in the revised version of the paper.
>
>  | Dataset | 10K | 5K | 1K |
>  | - | - | - | - |
>  | StarCoder | 10.7% | 11.8% | 6.8% |
>  | OpenWebMath | 6.9% | 4.5% | 8.0% |
>  | FineWeb-Edu | 63.8% | 65.9% | 64.6% |
>  | Wiki | 5.0% | 3.9% | 8.5% |
>  | Arxiv | 6.4% | 7.5% | 7.3% |
>  | Stackexchange | 5.0% | 1.0% | 2.7% |
>  | AlgebraicStack | 2.3% | 5.3% | 2.1% |
>
> Additionally, Figure 4 in the paper validates the effectiveness of the final reweighted mixture on downstream evaluation benchmarks. Despite that benchmarks are not used during any training or probing stage, the reweighted dataset consistently reduces perplexity on these held-out benchmarks relative to uniform sampling, further demonstrating the practical effectiveness of this influence-driven data selection approach with reasonable approximations.
>
> [1] AutoMixer: Checkpoint Artifacts as Automatic Data Mixers.
> [2] DataInf: Efficiently Estimating Data Influence in LoRA-tuned LLMs and Diffusion Models.
>
> [ **Definition of reasoning capabilities** ]
>
> Thanks for the insightful comment. We agree that “reasoning capability” is inherently multi-faceted, spanning planning, commonsense, multilingual reasoning, and more. We have shown the averaged accuracy on eight commonsense reasoning benchmarks including (HellaSwag, BoolQ, PIQA, SIQA, OBQA, ARC-challenge, ARC-easy, WinoGrande) in Table 8 of the paper. Other than that, we follow prior studies such as [3–6], to adopt performance on challenging math and coding benchmarks—which require multi-step derivations, intermediate symbolic manipulation, and explicit chain-of-thought reasoning—as a pragmatic and widely accepted proxy for assessing reasoning capability, allowing for controlled, reproducible comparisons.
>
> We acknowledge the value of broader coverage including planning, multilingual reasoning or even extend this reasoning to enable the agentic framework. Since this paper focuses on understanding how improved data curation can elicit stronger reasoning behaviors, which is illustrated through performance on challenging mathematical and coding tasks, delivering exhaustive reasoning capabilities across multiple domains is kind of beyond what we can achieve in a single study. We have added a discussion section in the revised paper to highlight these points as important directions for future research, and we hope our work can spark new ideas and provide a solid foundation for future explorations.
>
> [3] Chain-of-Thought Prompting Elicits Reasoning in Large Language Models.
> [4] Large Language Models are Zero-Shot Reasoners.
> [5] Qwen3 Technical Report.
> [6] DeepSeek-R1: Incentivizing Reasoning Capability in LLMs via Reinforcement Learning.

---

> ### Author Response · Authors · 2025-11-21
>
> [ **Compare with more data mixture baselines** ]
>
> Thank you for the questions. We hereby provide a conceptual comparison against both baselines including AutoMixer and In-Run Data Shapley, and additionally include an experimental comparison with AutoMixer.
>
> There are two key distinctions between AutoMixer and the data pipeline proposed in this paper: (1) AutoMixer regroups samples across datasets, blurring dataset boundaries, whereas our pipeline preserves datasets as-is, improving reproducibility and transparency. (2) AutoMixer computes influence scores directly using benchmark data, while X-LLM-R1 relies solely on probing datasets without accessing benchmarks during either training or probing.
>
> The table below reports accuracy results comparing AutoMixer and our X-LLM-R1 data-mixing method. We conducted controlled experiments by training an X-LLM-R1 950M model for 50K steps on 2 nodes under both data-mixing strategies. Despite not using benchmark data during sampling, X-LLM-R1 delivers performance comparable to AutoMixer on zero-shot commonsense reasoning, while attaining marginally higher accuracy on Math, GSM8K, and HumanEval benchmarks, demonstrating the effectiveness of influence estimation from probing datasets.
>
> | Model | CommonSense | Math | GSM8K | HumanEval |
> |-|-|-|-|-|
> | Automixer | 47.98 | 0.36 | 1.44 | 1.83 |
> | X-LLM-R1 Data Mix | 46.12 | 0.58 | 1.90 | 2.44 |
>
> Regarding the comparison with In-Run Data Shapley, which leverages gradients from simulated training runs to estimate each data source’s contribution, In-Run Data Shapley is theoretically more accurate because it captures the actual training dynamics. However, it lacks flexibility: the datamix ratio cannot be efficiently recalculated when the data composition changes. Any update requires rerunning the entire simulation. In contrast, our approach relies on a simpler influence-score aggregation mechanism, which avoids reruns and only requires renormalizing the existing influence scores when adding or removing some dataset in the training corpora.
>
> We have added these discussions to the related work section and add the experimental results in the appendix to make the paper more comprehensive.
>
> *Thank you once again for your insightful feedback, which has been instrumental in improving our work. We hope that our responses have resolved the questions you raised.*

---

### Official Review · Reviewer_GtCo · 2025-11-01

**Soundness:** 2
**Presentation:** 3
**Contribution:** 2
**Rating:** 6
**Confidence:** 4

**Summary:**

The paper presents a details of data-centric framework for creating high-performing sub-billion-parameter language models (LLMs) with strong reasoning capabilities, named X-LLM-R1. The central finding challenges the assumption that reasoning requires massive training corpora (>10T tokens) by demonstrating that models trained on only 4.2T curated tokens can match or exceed models trained on 36T proprietary tokens, like Qwen3-0.6B. Key methodological contributions include a benchmark-free, self-evolving data optimization approach using cross-domain influence scores to dynamically manage the training data mixture, along with a phased pre-training and mid-training curriculum focused on token efficiency. The authors highlight the critical role of data quality and structured post-training (SFT) over sheer scale, ultimately achieving state-of-the-art results among small, fully open-sourced reasoning models.

**Strengths:**

1. Novel data curation and training strategy

2. Challenging the conventional belief of training with large corpora of tokens for reasoning capability emergence

3. SoTA reasoning on SLMs, and making models open weight. The performance gain over other open source SLMs are convincingly better.

**Weaknesses:**

1. Inherent Limitations of Small Model Capacity still remains a questionable issue along with the constraint of performance on long context data.

2. Ineffectiveness of RL on SLMs is a concern here.

3. The concept of reasoning remains vague in the paper.

**Questions:**

1. Can you please test in details the long context inference performance with these models?

2. Please compare with SoTA reasoning models for the reasoning tasks (not only the instruction tuned models).

---

> ### Author Response · Authors · 2025-11-21
>
> Thank you for your positive feedback on our paper! We address each of your comments and questions as follows:
>
> [ **Inherent capacity limits of small LMs on long context performance** ]
>
> Thanks for raising this point. For the long-context reasoning performance, we evaluate our model on AIME’24 and LCBv6 benchmarks using a generation length of 32k, which is already a non-trivial context length. Within this range, X-LLM-R1-950M achieves comparable or even superior performance to Qwen3-0.6B, and approaches the results of several 1.5B-scale models. This demonstrates that the trained X-LLM-R1 has some capability in handling reasoning tasks that require long-context generation.
>
> Since our work specifically targets reasoning performance, we also would like to clarify that long-context tasks, such as needle-in-a-haystack [1] or retrieval/multi-hop tracing/aggregation tasks in the RULER [2] benchmark, are not the primary focus of this paper and are therefore out of scope. Targeting these datasets requires incorporating the corresponding long-context training data into the pipeline, which unfortunately cannot be accomplished within the limited timeframe of the rebuttal.
>
> While we admit there remains a natural performance gap between small and large models due to inherent capacity differences, this gap reflects a general trade-off between the model size and capability, rather than a weakness in our approach. We are actually working on some future work that specifically examines how to enhance small models’ long-context capabilities through the lens of error compounding and entropy. We believe this direction is promising, and we have incorporated a brief discussion of it into the future work section.
>
> [1] https://github.com/gkamradt/LLMTest_NeedleInAHaystack
> [2] RULER: What’s the real context size of your long-context language models?
>
>
> [ **Reinforce learning on small language models** ]
>
> Indeed whether using RL for training a small reasoning language model is widely discussed in previous papers as well. In general RL is not ineffective for SLM, but just not as effective as distilling from the larger teacher model. As shown in Appendix Fig. 10 (left), directly applying RL after the mid-training stage does improve SLM performance noticeably. However, given the availability of stronger large LLMs, we find that distilling from larger RL-trained teachers is a more efficient and stable strategy, i.e., using the larger teacher’s generated data for sequence-level distillation. As shown in Appendix Fig. 10 (right), the model trained after SFT with teacher generated data, achieves better performance than the model optimized from RL.
>
> This approach is also consistent with the design choices in recent state-of-the-art reasoning models such as Qwen3 and DeepSeek-R1. These work explicitly mentioned that they apply RL primarily to their largest models and then distill the reasoning behaviors into smaller variants, which is found to be more effective than training the small models directly with RL.
>
> [ **The concept of reasoning** ]
>
> We appreciate this insightful comment. Indeed, the concept of reasoning in LLMs remains broad and somewhat ambiguous in the literature. In this work, we follow prior studies such as [3-6], to adopt performance on challenging math and coding benchmarks—which require multi-step derivations, intermediate symbolic manipulation, and explicit chain-of-thought reasoning—as a pragmatic and widely accepted proxy for assessing reasoning capability. We also mentioned this choice in Section 2 (lines 86–91),
>
> > The notion of reasoning in large language models (LLMs) remains both complex and contested. While the term is sometimes used to describe a model’s ability to engage in structured, multi-step inference, it has also become a proxy for improved performance on challenging benchmarks. In this work, we adopt a pragmatic stance: we treat gains on reasoning-centric benchmarks as reasonable evidence of enhanced reasoning behaviors, while remaining cautious about equating such gains with genuine reasoning ability in the cognitive sense .
>
> We will expand our discussion of the reasoning concept, clarify why math and coding serve as pragmatic proxies for reasoning ability, and outline directions for studying small language models on broader reasoning tasks in the future work section.
>
>
> [3] Chain-of-Thought Prompting Elicits Reasoning in Large Language Models
> [4] Large Language Models are Zero-Shot Reasoners
> [5] Qwen3 Technical Report
> [6] DeepSeek-R1: Incentivizing Reasoning Capability in LLMs via Reinforcement Learning

---

> ### Author Response · Authors · 2025-11-21
>
> [ **Compare with SoTA reasoning models** ]
>
> We have indeed already compared our method against state-of-the-art reasoning models, including:
>
> - Qwen3-0.6B and Qwen3-1.7B: the small variants of Qwen series, which are the state-of-the-art reasoning LLMs, with the key innovation of having thinking and non-thinking modes.
>
> - OpenReasoning-Nemotron-1.5B: a reasoning model that is post-trained for reasoning about math, code and science solution generation.
>
> - DeepSeek-R1-Distill-Qwen-1.5B: a distilled reasoning model in the DeepSeek-R1 family.
>
> All of these baselines are reasoning-centric models, rather than purely instruction-tuned ones. Our model achieves competitive or superior performance across multiple reasoning benchmarks relative to these dedicated reasoning models, indicating that our comparisons already cover the strongest prior work in the current reasoning language model literature. And our model is the first with a fully open training recipe that matches the performance of these partially open-sourced, state-of-the-art reasoning models.
>
>
> *We hope our detailed responses have adequately addressed your concerns. If so, we would greatly appreciate your consideration in further increasing your evaluation score. Thank you again for your thoughtful feedback.*

---

### Author Response · Authors · 2025-11-21

We thank reviewers for their positive feedback and insightful comments. We are glad that reviewers recognized the following strengths:

- **Important research direction**: challenges the common “scaling is all you need” assumption and demonstrates that strong reasoning capability can emerge in small models through carefully optimized data recipes. [Reviewers **GtCo**, **3Hy6**, **dhHj**, ​​**9Hby**]

- **Novel and principled methodology**: utilizes novel “benchmark-free, self-evolving” optimization that avoid overfitting and ensure generalizability. [Reviewers **GtCo**, **9Hby**]

- **Strong experimental results**: significantly outperforms other fully open-source models. [Reviewers **GtCo**, **3Hy6**, **dhHj**, ​​**9Hby**]

- **Commendable transparency**: All training recipes, datasets, and checkpoints are fully open-sourced [Reviewers **3Hy6**]. This makes for an extremely valuable resource for others to build upon. [Reviewers **dhHj**, **9Hby**].

The discussion raised a lot of insightful and valuable questions that have helped us strengthen the comprehensiveness of the paper. In response to the reviewers’ comments, we have revised the manuscript accordingly, with all modifications highlighted in blue. The major modifications are listed below:

- Added a discussion section of the reasoning concept, and outlined directions for studying small language models on broader reasoning tasks in the future work section. [Reviewers **GtCo**, **3Hy6**]

- Included variance analysis of the sensitivity to capability-probing datasets. [Reviewers **3Hy6**, ​​**9Hby**]

- Compared with more datamix baselines. [Reviewers **3Hy6**]

- Added analyses comparing how strong reasoning-centric training impacts language understanding, factual retention, and general instruction alignment.  [Reviewers **dhHj**]

- Added ablations on more frequent reweighting and on whether it is necessary to compute the influence for new models. [Reviewers **dhHj**]

- Included the computational cost of data curation. [Reviewers **dhHj**, ​​**9Hby**]

- Listed the percentage of data that was filtered out in each stage. [Reviewers **9Hby**]

Overall, we appreciate the reviewers’ recognition of the paper’s new insights, as well as its openness and reproducibility. We hope these contributions can help drive continued progress toward fully open language models in the broader community.

---

### Meta-Review · Area_Chair_rZ5A · 2026-01-04

**Summary:**

The paper presents a data-centric framework for creating high-performing sub-billion-parameter LLMs with strong reasoning capabilities. The central finding challenges the assumption that reasoning requires massive training corpora (>10T tokens) by demonstrating that models trained on only 4.2T curated tokens can match or exceed models like Qwen3-0.6B. Key methodological contributions include a benchmark-free, self-evolving data-optimization approach that uses cross-domain influence scores to dynamically manage the training data mixture, along with a phased pre-training and mid-training curriculum focused on token efficiency. The study shows the critical role of data quality and structured post-training (SFT) over sheer scale, ultimately achieving state-of-the-art results among small, fully open-sourced reasoning models.

**Reviewer Concerns:**

Concerns addressed by the rebuttal:
- Comparison with state-of-the-art reasoning models (GtCo, 9Hby): The authors clarified that comparisons already include leading reasoning-centric baselines (Qwen3, DeepSeek-R1, Nemotron).
- Influence estimation reliability and variance (3Hy6): Hessian–vector product (HVP) approximation and variance analysis is shown by the authors.
- Missing baselines such as AutoMixer / Data Shapley (3Hy6): extra experiments regarding AutoMixer are provided, and the discussion on data shapley is also provided.
- Compute cost of data curation (dhHj, 9Hby): Detailed GPU-hour accounting shows curation overhead is ~15% of total cost.
- Knowledge vs. reasoning trade-offs (dhHj): Additional evaluations (RACE, TruthfulQA, IFeval) are provided.

Concerns partially or still outstanding:
- Long-context generalization (GtCo): While 32k-generation results are reported, broader long-context retrieval-style tasks remain unexplored.
- Breadth of reasoning capabilities (3Hy6): Planning, multilingual reasoning, and agentic behaviors are acknowledged but not empirically explored.
- Generality beyond current probing design (9Hby): Although sensitivity analyses help, full end-to-end retraining variance remains future work.

**Reviewer Scores:**

Reviewer GtCo and Reviewer 9Hby initially gave a score of 6 (marginally above the acceptance threshold). I think there is a chance that they would have changed their scores if they had been able to participate fully in the discussion.

---

### Decision · Program_Chairs · 2026-01-26

Accept (Poster)